

**Using observed urban NOₓ sinks to constrain VOC reactivity and the ozone and radical**
**budget in the Seoul Metropolitan Area**
Benjamin A. Nault[1,2,*], Katherine R. Travis[3], James H. Crawford[3], Donald R. Blake[4], Pedro
Campuzano-Jost[5], Ronald C. Cohen[6], Joshua P. DiGangi[3], Glenn S. Diskin[3], Samuel R. Hall[7], L.
Gregory Huey[8], Jose L. Jimenez[5], Kyung-Eun Kim[9], Young Ro Lee[8,a], Isobel J. Simpson[4], Kirk
Ullmann[7], Armin Wisthaler[10,11]
[1]CACC, Aerodyne Research, Inc., Billerica, MA, USA
[2]Department of Environmental Health and Engineering, Johns Hopkins University, Baltimore,
MD, USA
[3]NASA Langley Research Center, Hampton, VA, USA
[4]Department of Chemistry, University of California, Irvine, CA, USA
[5]CIRES and Department of Chemistry, University of Colorado, Boulder, CO, USA
[6]Department of Chemistry, University of California, Berkeley, CA, USA
[7]Atmospheric Chemistry Observations & Modeling Laboratory, NCAR, Boulder, CO, USA
[8]School of Earth & Atmospheric Sciences, Georgia Institute of Technology, Atlanta, GA, USA
[9]School of Environmental Sciences and Environmental Engineering, Gwangju Institute of Science
and Technology, Gwangju, South Korea
[10]University of Oslo, Oslo, Norway
[11]University of Innsbruck, Innsbruck, Austria
[a]Now at Division of Geological and Planetary Sciences, California Institute of Technology,
Pasadena, CA, USA
[*]Corresponding author:
Email: bnault@aerodyne.com, bnault1@jh.edu





**Abstract**
Ozone ($O_3$) is an important secondary pollutant that impacts air quality and human health. Eastern
Asia has high regional $O_3$ background due to the numerous sources and increasing and rapid
industrial growth, which impacts the Seoul Metropolitan Area (SMA). However, SMA has also
been experiencing increasing $O_3$ driven by decreasing $NO_x$ emissions, highlighting the role of
local, in-situ $O_3$ production on SMA. Here, comprehensive gas-phase measurements collected on
the NASA DC-8 during the NIER/NASA Korea United States-Air Quality (KORUS-AQ) study
are used to constrain the instantaneous $O_3$ production rate over the SMA. The observed $NO_x$
oxidized products support the importance of non-measured peroxy nitrates (PNs) in the $O_3$
chemistry in SMA, as they accounted for ~49% of the total PNs. Using the total measured PNs
($\Sigma$PNs) and alkyl and multifunctional nitrates ($\Sigma$ANs), unmeasured volatile organic compound
(VOC) reactivity (R(VOC)) is constrained and found to range from $1.4 - 2.1$ s$^{-1}$. Combining the
observationally constrained R(VOC) with the other measurements on the DC-8, the instantaneous
net $O_3$ production rate, which is as high as ~10 ppbv hr$^{-1}$, along with the important sinks of $O_3$ and
radical chemistry, are constrained. This analysis shows that $\Sigma$PNs play an important role in both
the sinks of $O_3$ and radical chemistry. Since $\Sigma$PNs are assumed to be in steady-state, the results
here highlight the role $\Sigma$PNs play in urban environments in reducing net $O_3$ production, but $\Sigma$PNs
can potentially lead to increased net $O_3$ production downwind due to their short lifetime (~1 hr).
The results provide guidance for future measurements to identify the missing R(VOCs) and $\Sigma$PNs
production.





**Short Summary**
Ozone ($O_3$) is a pollutant formed from the reactions of gases emitted from various sources. In
urban areas, the density of human activities can increase the $O_3$ formation rate ($P(O_3)$); thus, impact
air quality and health. Observations collected over Seoul, South Korea, are used to constrain $P(O_3)$.
A high local $P(O_3)$ was found; however, local $P(O_3)$ was partly reduced due to compounds typically
ignored. These observations also provide constraints for unmeasured compounds that will impact
$P(O_3)$.





## 1. Introduction

Representing global and urban tropospheric ozone ($O_3$) in chemical transport models (CTMs) is still challenging due to uncertainty in physical and chemical processes that control the $O_3$ budget (Archibald et al., 2020). One area of uncertainty is underestimated urban volatile organic compounds (VOCs) emissions (von Schneidemesser et al., 2023), which arise form a large number of sources, including some that are very hard to quantify (e.g., cooking and chemical product) (e.g., McDonald et al., 2018; Simpson et al., 2020). Intensive research is also ongoing as to why $O_3$ is increasing in recent years in urban areas, even with reductions in combustion emissions (e.g., Lyu et al., 2017; Colombi et al., 2023). This $O_3$ impacts the large populations in urban areas with harmful health effects, including premature mortality (e.g., Cohen et al., 2017).

Tropospheric $O_3$ production is driven by the catalytic cycling of nitrogen oxides ($NO_x = NO + NO_2$) fueled by the photoxidation of VOCs, both of which can come from anthropogenic emissions. The chemistry producing $O_3$ is described in R1 – R6 in Table 1. During daylight hours, VOCs are oxidized by OH (or undergo photolysis) to form an organic peroxy radical ($RO_2^{\cdot}$) in R1a (R1b). If the $RO_2^{\cdot}$ then proceeds through R2a, at least two $O_3$ molecules are produced. The first $O_3$ molecule is formed by the photolysis of $NO_2$ and the reaction of $O(^3P)$ with oxygen (R3 – R4). The second $O_3$ molecule is formed through the reaction of the alkoxy radical ($RO^{\cdot}$) with oxygen to form the hydroperoxyl radical ($HO_2$) (R5), which goes on to react with NO to produce $NO_2$ (R6) and the subsequent reactions described above (R3 – R4). However, some fraction of the time, depending on the number of carbons and functional group (e.g., Espada and Shepson, 2005; Perring et al., 2013; Yeh and Ziemann, 2014), alkyl or multifunctional nitrates ($ANs \equiv RONO_2$) are formed (R2b). The fraction of reactions to form ANs is described by the branching ratio, α. Reaction R2b has been shown to impact $O_3$ production, depending on the types of VOC emitted,



by reducing the fraction of $NO_2$ that photolyzes to form $O_3$ in source regions (R3 – R4) (Farmer et
al., 2011). As $\alpha$ is a function of the individual VOC's carbon backbone and functional group (e.g.,
Perring et al., 2013), any uncertainty related to primary VOC emissions and secondary chemistry
will directly impact the ability to describe urban $O_3$ production.

One important subclass of VOCs aldehydes (RCHO), which can either be directly emitted

or produced via photooxidation of VOCs (Mellouki et al., 2015; de Gouw et al., 2018; Yuan et al.,
2012; Wang et al., 2022). The photooxidation of the aldehyde (R7) in the presence of $NO_x$ can
either form acyl peroxy nitrates (R8, PNs = $R(O)O_2NO_2$) or an organic peroxy radical ($RO_2^{\cdot}$) (R9).
The competition between R8 to form PNs versus R9 to form $RO_2^{\cdot}$ depends on the NO-to-$NO_2$ ratio
(Nihill et al., 2021). Further, R8 is in thermodynamic equilibrium due to the weak bond strength
between the acyl peroxy radical ($R(O)O_2^{\cdot}$) and $NO_2$. Thus, formation of PNs pose only a temporary
loss of $NO_2$. Finally, it has been observed that aldehydes with longer carbon backbones (e.g., C8s
and C9s) from various anthropogenic activities, such as cooking (Coggon et al., 2024; Rao et al.,
2010), may have mixing ratios as high as aldehydes typically quantified in field experiments
(acetaldehyde and propaldehyde). However, there is larger uncertainty associated with these higher
aldehydes in their fate to produce both PNs and ANs (e.g., Hurst Bowman et al., 2003). Missing
both these emissions and subsequent chemistry would impact estimates of urban $O_3$ chemistry.

The fraction of $RO_2^{\cdot}$ forming ANs in R2b and the fraction of $R(O)O_2^{\cdot}$ forming PNs in R8

alter the instantaneous $O_3$ production ($P(O_3)$) by removing $NO_2$ and/or the radical species. This is
further shown in Figure S1, where an analytical equation to describe R1 – R6 (Farmer et al., 2011),
is used to explore how changes in the VOC reactivity (R(VOC)), radical production ($P(HO_x)$), and
ANs production and branching ratio, $\alpha$ (R2b), impact the instantaneous $P(O_3)$ (see Sect. S1 for the
analytical equation and description). Any changes in $P(HO_x)$, R(VOC), and/or $\alpha$ will impact both



the instantaneous $P(O_3)$ as well as the $NO_x$ mixing ratio corresponding to the maximum $P(O_3)$. As
these parameters are generally interconnected, investigating all three is important to understand
the sources and control of instantaneous $P(O_3)$. Further, R7 – R9 are not included in this traditional
description of the analytical equation, as it is assumed PNs are in steady-state (Farmer et al., 2011).
Thus, if PNs are not in steady-state, their role in altering $P(O_3)$ may be underestimated.

Increasing surface $O_3$ is a concern throughout East Asia, including South Korea (Colombi

et al., 2023; Gaudel et al., 2018; Kim et al., 2021; Yeo and Kim, 2021). The emissions associated
with industry and other anthropogenic activities and the associated photochemistry have impacted
regional air quality, leading to high $O_3$ backgrounds that can impact a country's ability to achieve
reduced $O_3$ exposure for new air quality standards (e.g., Colombi et al., 2023). However, local
emissions and photochemistry still play an important role. For example, during the Korea-United
States Air Quality (KORUS-AQ) campaign, it was observed between morning and afternoon in
the Seoul Metropolitan Area (SMA), $O_3$ increased by ~20 parts per billion by volume (ppbv) over
a background concentration of over 75 ppbv (Crawford et al., 2021). Thus, an understanding of
the variables highlighted in Figure S1 are necessary to control both local and regional $P(O_3)$.

One tool typically used to understand the role of regional $O_3$ and transported $O_3$ on local

$O_3$ and impacts of local emission controls on $O_3$ are CTMs. As shown in Park et al. (2021), for the
SMA, CTMs typically underestimate the observed $O_3$ and formaldehyde. While the low $O_3$ could
be partially related to underestimated transport (e.g., Seo et al., 2018) or resolution of the CTM
(e.g., Jo et al., 2023; Park et al., 2021), the low bias also observed for modeled formaldehyde
indicates overall (a) too little VOCs and thus too low R(VOC) (Brune et al., 2022; H. Kim et al.,
2022), (b) missing photochemical products from missing VOCs, including oxygenated VOCs
(OVOCs) that contribute to $P(HO_x)$ (Brune et al., 2022; H. Kim et al., 2022; Lee et al., 2022; Wang





et al., 2022), and (c) likely missing PNs and ANs from the underestimated VOCs related to the
underestimated R(VOC) (Lee et al., 2022; Park et al., 2021). Missing (a) – (c) will bias the
instantaneous P($O_3$) (Figure S1), impacting the ability to investigate what policies should be
implemented to reduce $O_3$.

To better understand what controls the instantaneous P($O_3$) over SMA, observations

collected on the NASA DC-8 during KORUS-AQ are used to constrain the three variables
highlighted in Figure S1—R(VOC), $HO_x$ production and loss, and ANs and PNs production.
Observational constraints on these three parameters provide a means to investigate the
instantaneous P($O_3$) over SMA and the major classes of contributors to $O_3$ and $HO_x$ production
and loss. These results are discussed and placed into the context of improving our knowledge about
$O_3$ production in an urban environment.

**2. Methods and Data Description**
**2.1 KORUS-AQ and DC-8 Descriptions**

The KORUS-AQ campaign was a multi-national project that was conducted in May – June,

2016, led by South Korea's National Institute of Environmental Research (NIER) and United
States National Aeronautics and Space Administration (NASA). The project was conducted in
South Korea and the surrounding seas with numerous airborne platforms, research vessels, and
ground sites (Crawford et al., 2021). The study here focuses on the observations collected on the
NASA DC-8.

The instrument payload, flights, and observations have been described in other studies

(Crawford et al., 2021; Schroeder et al., 2020; Brune et al., 2022; Lee et al., 2022). Briefly, the
DC-8 was stationed at Osan Air Force Base, Pyeongtaek, South Korea, which is approximately 60



km south of Seoul. A total of 20 research flights were conducted with the DC-8. Part of each
research flights included a stereo-route in the SMA in the morning (~09:00 local time), midday
(~12:00 local time), and afternoon (~15:00 local time), which included a missed approach over
Seoul Air Base (< 15 km from Seoul city center) and a fly-over of the Olympic Park and Taehwa
Forest Research sites (Figure 1). A total of 55 descents over Olympic Park and 53 spirals over
Taehwa Forest Research site were conducted (Crawford et al., 2021). Only observations from the
DC-8 after 11:00 local time are used here to ensure that the boundary layer has grown and
stabilized and to minimize any influence from residual layer mixing into the boundary layer and/or
titration of $O_3$ by NO (R10). We analyze data collected below 2 km and between $127.10 – 127.67°E$
and $37.22 – 37.69°N$ to focus on the boundary layer in the SMA without influence from industrial
emissions along the western South Korean coast (Crawford et al., 2021).

During KORUS-AQ, four different meteorological periods, as described by Peterson et al.

(2019), impacted the region. These periods included a Dynamic period from $1 – 16$ May, where
there were a series of frontal passages; a Stagnant period from $17 – 22$ May, where it was dry,
clear, and stagnant; Transport/Haze period from $25 – 31$ May, where long-range transport and hazy
conditions with high humidity and cloud cover prevailed; and, a Blocking period from $1 – 7$ June,
where blocking conditions minimized transport (Peterson et al., 2019). However, as discussed in
Sect. 3.2, conditions did not impact the general trends and chemistry and thus the whole campaign
has been analyzed together.

The observations used for the analysis are shown in Table 2, along with the associated

references. The 1-min merged data from the DC-8 is used here (KORUS-AQ Science Team, 2023).
For data missing due to frequency of measurements (e.g., VOCs from WAS), data was filled in a
similar approach as Schroeder et al. (2020), in that VOCs with missing data were filled by the





linear relationship of that VOC with VOCs measured more frequently. This step was necessary for
the observations used in the diel steady-state calculations described in Sect. 2.2. Note, the TD-LIF
$NO_2$ (see Table 2) was used throughout this study and discussed in Sect. S2 and Figure S2 – S3 as
it generally agreed better with steady-state calculated $NO_2$-to-NO ratios than the
chemiluminescence $NO_2$.

**2.2 F0AM Box Model Diel Steady-State Calculations for Missing Reactivity and**
**Peroxynitrate Budget Analysis**

We use the F0AM box model (Wolfe et al., 2016) with chemistry from the MCMv3.3.1

(Jenkin et al., 2015) to simulate production of PNs and formaldehyde using 1-min merged data, as
described in Sect. 2.1. As in Schroeder et al. (2020), we simulate each aircraft observation in
diurnal cycle mode until the diurnal cycle for each species reaches convergence within 1%. We
constrain concentrations of NO, $O_3$, $H_2O_2$, $HNO_3$, CO, $CH_4$, $H_2$, and all measured or estimated
VOCs given in Table 2 and Table S1. We allow the model to freely calculate $NO_2$, formaldehyde,
and all PNs, including PAN and PPN, for when calculating the budget of PNs. However, for the
acyl peroxy radical mixing ratios to calculate $O_x$ and $HO_x$ budget (Sect. 2.3), PAN and PPN were
constrained by observations. We use a dilution constant of 12 hours, according to Brune et al.
(2022). Model evaluation is discussed in Sect. 3.4. The contribution of individual VOCs to PAN
was calculated by reducing precursor VOCs by 20% and multiplying the resulting impact on the
peroxy acetyl radical ($CH_3C(O)O_2^.$) by 5. Other acyl peroxy nitrates (higher PNs) are lumped into
categories based on their primary precursor species from Table S2, species currently typically



measured (e.g., PPN) or contributes a large fraction of the total higher PNs budget (greater than
>2%; e.g., PHAN and MPAN).

**196    2.3 Calculation of Instantaneous Ozone and $HO_x$ Production and Loss**

An experimental budget for the production and loss of $O_x$ ($O_x = O_3 + NO_2$) and $HO_x$ ($HO_x$
$= OH + HO_2 + RO_2^{\cdot} + R(O)O_2^{\cdot}$) is described here. $NO_2$ and $O_3$ are combined to reduce any potential
impact from titration via $O_3$ reaction with NO (R10). The budget analysis includes field-measured
quantities (mixing ratios and photolysis rates, Table 2), results from F0AM (Sect. 2.2), estimated
missing R(VOC) (Sect. 3.2) and published kinetic rate constants (see Table 1 for references). The
rate of production or destruction is calculated with the following equations (Eq. 1 – 7) below. Note,
these equations differ from Schroeder et al. (2020) in that (a) ANs and PNs chemistry are explicitly
included and (b) the reaction of $O_3$ with alkenes is excluded as this reaction contributed a minor
loss to $O_3$ (< 1%).

$P_{O_x} = \sum_i (1-\alpha_{eff}) k_{RO_{2,i}+NO}[RO_{2,i}^{\cdot}][NO] + k_{HO_2+NO}[HO_2][NO]$      (1)

$L_{O_x} = k_{NO_2+OH}[NO_2][OH] + k_{O_3+OH}[O_3][OH] + f \times j_{O^1D}[O_3] +$

$k_{HO_2+O_3}[HO_2][O_3] + net(PNs)$      (2)

$net(PNs) = \beta k_{R(O)O_2^{\cdot}+NO_2}[R(O)O_2^{\cdot}][NO_2] - (1-\beta) k_{decomposition}[PNs]$      (3)

$\beta = \dfrac{k_{RC(O)O_2+NO_2}[NO_2]}{k_{RC(O)O_2+NO_2}[NO_2] + k_{RC(O)O_2+NO}[NO]}$      (4)

$P(HO_x) = 2f \times j_{O^1D}[O_3] + 2j_{H_2O_2}[H_2O_2] + 2j_{CH_2O \rightarrow H+HCO}[CH_2O] + 2j_{CHOCHO}[CHOCHO] +$

$2j_{CH_3OOH}[CH_3OOH] + 2j_{CH_3CHO}[CH_3CHO] + 2j_{CH_3C(O)CH_3}[CH_3C(O)CH_3] +$
$2j_{CH_3CH_2C(O)CH_3}[CH_3CH_2C(O)CH_2]$      (5)



$$L(HO_x) = k_{NO_2+OH}[NO_2][OH] + \sum_i \alpha_{eff} k_{RO_{2,i}+NO}[RO_{2,i}^{\cdot}][NO] +$$
$$2k_{HO_2+HO_2}[HO_2][HO_2] + 2k_{RO_2+RO_2}[RO_2^{\cdot}][RO_2^{\cdot}] + 2k_{HO_2+RO_2}[HO_2][RO_2^{\cdot}] + net(PNs) \quad (6)$$
$$[RO_2^{\cdot}] = \frac{\sum_i k_{OH+VOC,i}[VOC_i][OH]}{(1-\alpha_{eff})k_{RO_2+NO}[NO]+k_{RO_2+HO_2}[HO_2]} \quad (7)$$
Here, $k$ is the rate constant for compound, $i$, with the associated compound listed, $\alpha_{eff}$ is the
effective branching ratio for R2a and R2b for the observations (Sect. 3.2), $f$ is the fraction that $O^1D$
that reacts with water to form OH versus reacting with a third body molecule to form $O^3P$, $\beta$ is the
fraction the $R(O)O_2^{\cdot}$ that reacts with $NO_2$ versus NO, and $j$ is the measured photolysis frequency
(Table 2). Note, $R(O)O_2^{\cdot}$ is not included in Eq. 7 as (a) it is assumed the initial production of
$R(O)O_2^{\cdot}$ is captured with the reaction of OH with VOC and (b) $R(O)O_2^{\cdot}$ accounts for a small
fraction of the total $RO_2$ (< 10%). Not including $R(O)O_2^{\cdot}$ in Eq. 7 may lead to a small
underestimation of total $RO_2^{\cdot}$. Finally, $HO_2$ calculated from F0AM is used in the equations to
determine the $O_x$ and $HO_x$ budget.

**3. Observational constraints on $NO_x$ organic oxidation chemistry**
In the Sect. 3.1, the detailed observations from the DC-8 during KORUS-AQ provided
measurements that allow us to test our understanding of $NO_x$ oxidation into total $NO_z$ ($NO_z$ =
higher $NO_x$ oxides, including $\Sigma$PNs, $\Sigma$ANs, $HNO_3$ and particulate nitrate, $pNO_3$), which is needed
for the remainder of the analysis. Sect. 3.2 to 3.4 will focus on the organic $NO_z$ chemistry. This is
due to the chemistry and dynamics impacting the total inorganic nitrate chemistry that has been
discussed recently (Travis et al., 2022; Jordan et al., 2020).

**3.1 $NO_x$ and its oxidation products**



The average NO$_x$ mixing ratios observed by the NASA DC-8 in the SMA below 2 km after
11:00 local time is shown in Figure 1. As NO$_x$ is mainly emitted from anthropogenic activities,
such as combustion emissions, in an urban environment, the largest NO$_x$ mixing ratios are
observed between Olympic Park and the missed approach, as this area included downtown SMA.
As the DC-8 flies from the missed approach toward Taehwa Research Site, the NO$_x$ mixing ratios
decreases. The combination of reduced emissions, chemical reactions, and dilution and mixing
reduces the NO$_x$ mixing ratios away from the city. An understanding of these processes is
important for urban P(O$_x$).
On the DC-8, there were multiple measurements of various speciated and total family
contribution towards NO$_z$ (Table 2). The comparison of the speciated and measured NO$_z$ is
investigated in Figure 2 for observations over SMA. When only speciated PNs (GT) and ANs (CIT
+ WAS) and gas-phase nitrate (HNO$_3$) are compared to the NO$_z$ (NO$_y$ (NCAR) – (NO (NCAR) +
NO$_2$ (TD-LIF)), only 46% of the NO$_z$ can be explained. This is not completely unexpected, as
multiple studies have indicated that the speciated ANs measurements are typically lower than the
total ANs measurements (Perring et al., 2010; Fisher et al., 2016). Further, pNO$_3$ has been found
to be important for total nitrate budget in the SMA (e.g., Travis et al., 2022). Chemiluminescence
measurements of gas-phase NO$_y$ have been found to efficiently measure pNO$_3$, depending on the
sensitivity to pNO$_3$ enhancements or exclusions (Bourgeois et al., 2022); thus, it is expected that
missing ANs and pNO$_3$ are necessary to close the NO$_z$ budget. Adding the measured pNO$_3$ to the
speciated PNs (GT) and ANs (CIT + WAS) and gas-phase nitric acid, 81% of NO$_z$ can be
explained. This barely overlaps the combined uncertainty of the measurements (~26%). Total PNs
and ANs, measured by TD-LIF, are needed to close of the total NO$_z$ budget.





The breakdown of the $NO_z$ budget over the SMA as the airmasses photochemically ages
(decreasing $NO_x$ contribution to total $NO_y$) is shown in Figure 2b. During KORUS-AQ, ~56% of
$NO_z$ was inorganic (gas- and particle-phase nitrate), ranging from 52% to 62%; the remaining $NO_z$
was organic (PNs and ANs). Approximately 74% of the total ANs were not speciated (range 73%
to 76%). Speciated PNs species, such as PAN (peroxy acetyl nitrate), account for a mean 51% of
the total PNs (range 47 to 59%), much lower than typically observed in prior studies (e.g.,
Wooldridge et al., 2010). In these prior studies, the speciated PN species (typically PAN + PPN
(peroxy propionyl nitrate)) accounted for 90 – 100% of the $\Sigma$PNs, except for some select cases
attributed to poor inlet design (Wooldridge et al., 2010). PAN accounted for the majority of the
speciated PNs, with the remaining speciated PNs (PPN + PBzN (peroxy benzoyl nitrate) + APAN
(peroxy acryloyl nitrate)) accounting for ~1%. However, during KORUS-AQ, Lee et al. (2022)
observed that PAN contributed only 60% of calculated total PNs in industrial plumes near the
SMA. Thus, the VOC emissions in and near SMA potentially lead to PNs typically not directly
measured; this is explored more in Sect. 3.4
As $NO_x$ decreases from ~30 ppbv to 4 ppbv, the contribution of organic $NO_z$ increases
(Figure 2b). At about 4 ppbv, the contribution of organic $NO_z$ starts to decrease. Further, the
contribution of the different organic $NO_z$ species changes. For example, from ~30 ppbv to 4 ppbv,
the un-speciated $\Sigma$PNs contributes the majority of the organic $NO_z$ budget (~39%). Below ~4 ppbv,
the contribution of un-speciated $\Sigma$PNs decreases and the PAN contribution increases. The change
in contribution of PNs is due to changes in the PN precursors (e.g., combination short-lived
precursors oxidizing to $CH_3C(O)O_2^{\cdot}$ and thermal decomposition of the higher PNs (higher PNs =
$\Sigma$PNs – PAN)). On the other hand, the contribution of un-speciated $\Sigma$ANs remains relatively
constant with $NO_x$ (~6% of total $NO_z$). However, the type of ANs is most likely changing with



NO$_x$ due to the lifetime of the ANs precursors and/or the lifetime of ANs. Less is known about the
lifetime of ANs derived from anthropogenically emitted VOCs compared to those from biogenic
VOCs (González-Sánchez et al., 2023; Picquet-Varrault et al., 2020; Zare et al., 2018). On average
unknown ANs and PNs account for ~24% of the observed NO$_z$ on average.

**3.2 Meteorological impact on NO$_x$ oxidation**

As discussed in Sect. 2.1 and various prior studies, four different meteorological conditions

impacted the observations during KORUS-AQ (Peterson et al., 2019). The impact of the
meteorological conditions on NO$_x$ oxidation was investigated by plotting two metrics of NO$_x$
oxidation—O$_x$ versus ΣANs and ΣPNs versus formaldehyde (Figure 3). The implications of both
plots are further discussed in Sect. 3.3 and 3.4, respectively. Briefly, O$_x$ versus ΣANs and ΣPNs
versus formaldehyde are competitive products from the reaction of RO$_2^\cdot$ or R(O)O$_2^\cdot$ with NO$_x$
(R2a versus R2b or R8 versus R9). The different meteorological periods corresponded to
differences in temperatures and amount of photolysis due to cloud cover (Peterson et al., 2019).
Thus, these different periods may impact gas-phase chemistry and/or VOC emissions. However,
as demonstrated in Figure 3, there are minimal systematic differences in the trends observed for
the two NO$_x$ oxidation products as there is no systematic shift in the trends or scatter observed in
Figure 3. This suggests that the data does not have to be separated by meteorological conditions.

**3.3 Production of ANs to constrain R(VOC)**

Observations of un-speciated ANs and PNs imply missing VOCs that impact O$_3$ chemistry.

The relationship of ANs to O$_x$ can provide a method to investigate this source. This relationship
provides an estimate of the effective branching ratio, α, for the observed VOC mix (Perring et al.,





2013 and references therein). The value of this relationship stems from the reactions discussed
above (R1 – R6) in that upon the oxidation of VOCs, some fraction of the time, $RO_2^.$ reacts with
NO to form an AN molecule and the remainder of the time the reaction goes to form $O_3$. This is
expressed with the following equations:
$$P_{\Sigma ANs} = \sum \alpha_i\, k_{OH+VOCi}[OH][VOC_i] \qquad (8)$$
$$P(O_x) = \sum_i \gamma_i(1 - \alpha_i)\, k_{OH+VOC_i}[OH][VOC_i] \qquad (9)$$
Here, α is the effective branching ratio in the reaction of $RO_2^.$ with NO to form ANs versus $RO^.$
(R2), $k$ is the OH rate constant with VOC, i, and γ is the number of $O_3$ molecules formed per
oxidation of VOC, i. The γ, calculated for the observed and calculated compounds from F0AM
using the values from MCM (Jenkin et al., 2015), is found to be, on average, 1.53, which is lower
than the value of 2 typically assumed in prior studies (e.g., Perring et al., 2013). This lower γ is
due to the role of CO and $CH_2O$ to the total reactivity. After the boundary layer height has
stabilized (e.g., after 11:00 am LT used here) and is near enough (e.g., less than 1 day aging) to
the VOC source to ignore deposition and entrainment, Eq. 8 and 9 can be combined to approximate
the change in $O_x$ per molecule ΣAN formed:
$$\frac{\Delta O_x}{\Delta \Sigma ANs} \approx \frac{P_{O_x}}{P_{\Sigma ANs}} \approx \frac{1.53(1-\alpha)}{\alpha} \qquad (10)$$
For this equation to be valid, α needs to be relatively small (α << 1), which is true for VOCs, as
maximum α for the conditions of KORUS-AQ is expected to be 0.35 (Orlando and Tyndall, 2012;
Perring et al., 2013; Yeh and Ziemann, 2014). Note, though Eq. 10 can be used at short
photochemical ages due to minimal impact from physical loss processes, chemical loss processes
may impact the assumptions in Eq. 10 and are discussed in more detail below.

Over the SMA during KORUS-AQ, the slope between $O_x$ and ΣANs was observed to be

40.5±1.8 (Figure 3a), with an $R^2$ = 0.60. Using Eq. 10, this translates to an effective branching



ratio ($\alpha_{eff}$), of 0.036. For other urban locations around the world, this slope has ranged from 13 –
47 (Farmer et al., 2011; Kenagy et al., 2020; Perring et al., 2010; Rosen et al., 2004), leading to an
effective $\alpha$ between 0.04 and 0.15, assuming a $\gamma$ of 2 instead of the calculated $\gamma$ used here. Thus,
the $\alpha_{eff}$ observed over SMA during KORUS-AQ is similar to other urban locations (Houston =
0.05 (Rosen et al., 2004) and South Korea = 0.05 (Kenagy et al., 2021)) but much lower than
observed for Mexico City = 0.07 – 0.12 (Perring et al., 2010; Farmer et al., 2011) and Denver =
0.16 (Kenagy et al., 2020). This suggests that VOCs with low $\alpha$ dominate the total R(VOC) and
production of ANs in SMA. The VOCs in SMA that dominate R(VOCs), including OVOCs,
alkenes, and aromatics (Schroeder et al., 2020; Simpson et al., 2020), generally have lower $\alpha$
(Perring et al., 2013 and references therein; Orlando and Tyndall, 2012).

We use the observed VOCs (Table 2) to calculate $\alpha_{eff}$ from this mixture to compare to the

calculated $\alpha_{eff}$ of 0.036 derived from the slope of $O_x$ versus $\Sigma$ANs in Figure 3a, as shown in Figure
4. The R(VOC) calculated from the observed VOCs and from the intermediates produced by the
F0AM model, described in Sect. 2.2, are shown in Figure 4a, and the reactivity weighted $\alpha$ for the
observations is shown in Figure 4b. As has been observed in other urban environments (e.g.,
Hansen et al., 2021; Whalley et al., 2016; Whalley et al., 2021; Yang et al., 2022;), measured
OVOCs contribute the most to the calculated R(VOC) for all $NO_x$ mixing ratios (32 – 48%). The
unmeasured OVOCs (F0AM species) contributed 17 – 28% of the calculated reactivity. The F0AM
species reactivity ranged from 0.45 – 1.78 $s^{-1}$, which is a similar increase in total OH reactivity
observed by Brune et al. (2022) over South Korea. At higher $NO_x$ mixing ratios, primary, more
reactive VOCs (e.g., alkanes, alkenes, aromatics) contribute an important fraction (> 25%) of the
R(VOC). As there are interferences in the total OH reactivity measurement at high $NO_x$ (Brune et
al., 2022), we are unable to determine the extent to which the observed and modeled reactivity



captures total OH reactivity in the SMA above a $NO_x$ value of approximately 4 ppbv. At lower
$NO_x$ mixing ratios, ~33% of the R(VOC) is missing (calculated R(VOC), including F0AM species,
~3.0 $s^{-1}$ and measured R(VOC) from Penn State—see Table 2—is 4.5 $s^{-1}$).

Numerous other urban studies have observed unmeasured OH reactivity, which is assumed

to be unmeasured R(VOC), as the inorganic OH reactivity is typically well covered by
measurements. This unmeasured R(VOC) has ranged from ~3 $s^{-1}$ to ~10 $s^{-1}$ (e.g., Brune et al.,
2022; Hansen et al., 2021; Kim et al., 2016; Ma et al., 2022; Tan et al., 2019; Whalley et al., 2016;
Whalley et al., 2021). Over the SMA, the difference between measured and calculated R(VOC)
was ~1.5 $s^{-1}$ at low $NO_x$ and unknown at high $NO_x$ mixing ratios. The lower difference may be
related to the comparison occurring for observations at low $NO_x$, when the very reactive material
has either reacted into compounds measured on the DC-8 (e.g., formaldehyde, acetaldehyde, etc.),
diluted to low enough concentrations to be negligible for R(VOC), or undergone deposition or
partitioning to the particle-phase.

At higher $NO_x$ mixing ratios, which is more representative of fresh emissions, these more

reactive compounds typically not measured are expected to lead to a higher difference between the
calculated and observed R(VOC). Prior studies with more comprehensive measurements found
these more reactive compounds and their secondary products contributed an important fraction
towards the R(VOC) (e.g., Whalley et al., 2016). Thus, to determine if these unmeasured VOCs
potentially contribute to the R(VOC), and thus $P(O_x)$, in SMA, another means to constrain their
contributions is necessary. One potential means to constrain the total R(VOC) is by using the
observed ΣANs and $O_x$ and assuming the observations are from the instantaneous production of
both species (e.g., the assumption used for Figure 3a).



To estimate the unmeasured R(VOC), Eq. 10 is used without cancelling out terms and
expanded into the measured and unmeasured R(VOC) and α:
$$\frac{\Delta O_x}{\Delta \sum ANs} = \frac{\gamma RVOC_m[OH]+\gamma RVOC_u[OH]-\gamma\alpha_m RVOC_m[OH]-\gamma\alpha_u RVOC_u[OH]}{\alpha_m RVOC_m[OH]+\alpha_u RVOC_u[OH]}$$     (11)
Here, $\frac{\Delta O_x}{\Delta \sum ANs}$ is the slope from Figure 3a, γ is the number of $O_3$ molecules formed per oxidation of
VOC, which is 1.53 for this study, R(VOC) is the VOC reactivity, which is its OH oxidation rate
constant and its concentration (k×[VOC]) in units $s^{-1}$, α is the branching ratio for R2 (Table 1), and
*m* and *u* correspond to measured and unmeasured RVOC and α. The rate constants for the measured
VOCs are listed in Table 1, the reactivity for F0AM is taken directly from F0AM, and α is either
from MCM (Jenkin et al., 2015) or Perring et al. (2013) for observations or assumed to be 0.05 for
F0AM secondary products. The equation is rearranged and solved for $RVOC_u$, using different
values of $\alpha_u$ (e.g., 0.00 – 0.30, values typical α).
As discussed in Sect. S3 in the Supp. Information, there are numerous assumptions and
potential sources of uncertainty in the simplified version of Eq. 11. A thorough analysis and
discussion of these assumptions are discussed in Sect. S3. The potentially most important
assumption is that chemical loss is negligible in solving Eq. 11. However, due to the expected
relatively short lifetime of ∑ANs, the chemical loss of both $O_x$ and ANs nearly cancel each other,
leading to similar results in considering or neglecting these loss terms in Eq. 11. Further, as ∑ANs
chemical loss has uncertainty, especially for ANs produced from anthropogenic VOC oxidation,
the use of Eq. 11 reduces some of these uncertainties in comparison to Eq. S9. Thus, for the
remainer of the paper, the values calculated from Eq. 11 will be used.
For the range of missing α assumed, an α = 0.10 for the unmeasured R(VOC) provides the
best agreement with the observed R(VOC) ("From PSU" is the Penn State OH Reactivity with
inorganic reactivity subtracted out) for all observations where $NO_x < 4$ ppbv. Further, it is found





that α ranging from 0.075 – 0.125 encompasses the associated uncertainty with the observed
R(VOC) ($\pm0.64$ s$^{-1}$ (Brune et al., 2019)). This leads to an average unmeasured R(VOC) of $1.7^{+1.1}_{-0.4}$.

The associated total missing R(VOC) for the assumed α of 0.10 ranges from 1.4 to 2.1 s$^{-1}$.

Assuming typical rate constants for emitted VOCs, assuming it is comparable to semi- and
intermediate-VOCs, and their associated secondary products ($\sim1 - 4\times10^{-11}$ cm$^3$ molec.$^{-1}$ s$^{-1}$ (Ma et
al., 2017; Zhao et al., 2014)), the total missing reactivity would be equivalent to $\sim1 - 8$ ppbv. Zhao
et al. (2014) observed $\sim12$ µg m$^{-3}$ of semi- and intermediate-VOCs near Los Angeles, CA, during
the CalNex study. Depending on the molecular weight assumed, this translates to $\sim1$ to 2 ppbv.
Nault et al. (2018) found that $\sim5 - 8$ ppbv of VOCs were needed to explain the observed secondary
organic aerosol production over the SMA, depending on the molecular weight assumed for the
VOC. Further, Kenagy et al. (2021) also found that known chemistry could only account for $\sim33\%$
of the observed ANs and missing sources of lower volatility VOCs to produce anthropogenically-
derived ANs were necessary. Finally, Whalley et al. (2016) found that addition of unassigned
VOCs and their associated oxidation products led to a reactivity of $\sim1.6$ s$^{-1}$, leading to $\sim1 - 6$ ppbv
missing R(VOC). Thus, the reactivity and equivalent mixing ratios estimated here appear plausible
and warrant future measurements to understand this unmeasured reactivity sources.

One important aspect of this unmeasured R(VOC) is that it should not be considered one

or a couple of VOCs emitted and contributing $1 - 8$ ppbv of VOC in the atmosphere. Instead, it
will be the emitted VOCs and its oxidation products summed together to form the $1 - 8$ ppbv of
unmeasured VOCs in the atmosphere.

One possible missing VOC is nonanal, which is associated with cooking emissions (Rao et

al., 2010; Sai et al., 2012; Schauer et al., 2002) and vegetative emissions (Hurst Bowman et al.,
2003). Kim et al. (2018) observed cooking organic aerosols at a ground site in SMA, indicating



that there should be associated gas-phase emissions from cooking. Nonanal has recently been
suggested to be a potential interference compound with isoprene measurements on a PTR-MS
(Coggon et al., 2024; Wargocki et al., 2023). Comparisons of isoprene measured by the PTR-MS
and WAS during KORUS-AQ (Figure S5) shows at increasing $NO_x$ mixing ratios (closer to
emission sources), the difference between the PTR-MS and WAS isoprene mixing ratios increases.
This suggests that there are potential unmeasured OVOCs and/or other $C_5H_8$ alkenes at high $NO_x$
ratios that cannot be easily determined by the difference between the PTR-MS and WAS.
Continuing to use nonanal as a surrogate for this unmeasured OVOC, nonanal has a rate constant
consistent with the values used above for the missing R(VOC) ($3.6 \times 10^{-11}$ $cm^3$ molec.$^{-1}$ s$^{-1}$ (Hurst
Bowman et al., 2003)). Further, nonanal has an estimated high $\alpha$ of ~0.2 (Hurst Bowman et al.,
2003). As typical nonanal mixing ratios have been observed or estimated to be < 500 pptv, this
suggests that nonanal or similar OVOCs may contribute to some of the missing reactivity (< 0.45
s$^{-1}$). Finally, nonanal may be an important higher PNs precursor (see Sect. 3.4 for more discussion
about un-speciated higher PNs).

OVOC emissions are generally considered to be an important fraction of R(VOC) for urban

emissions (de Gouw et al., 2018; Gkatzelis et al., 2021; McDonald et al., 2018; Ma et al., 2022;
Simpson et al., 2020; Wang et al., 2022; Yang et al., 2022). However, the $\alpha$ for OVOC is
potentially smaller than alkanes, though it is highly unconstrained (Orlando and Tyndall, 2012).
Note, higher OVOCs have been understudied and thus may have higher $\alpha$ (e.g., nonanal). Thus, if
the missing reactivity is mainly OVOCs and it is assumed their $\alpha$ is low, compounds with $\alpha > 0.15$
will be needed for the budget closure shown here. Likely compounds with high $\alpha$ include alkanes,
cycloalkenes/alkenes, and aromatics, though the latter is also highly uncertain. Alkanes have
typically been a small source for the R(VOC) in urban environments (e.g., McDonald et al., 2018;



Simpson et al., 2020; Whalley et al., 2016). Though aromatics contribute a significant fraction of
R(VOC) in different Asian urban environments (Brune et al., 2022; Schroeder et al., 2020;
Simpson et al., 2020; Whalley et al., 2021), the majority of the aromatic R(VOC) is considered to
be measured by WAS over SMA during KORUS-AQ (e.g., measured aromatics account for ~81%
of aromatic reactivity in McDonald et al. (2018) and 98% of aromatic reactivity in Whalley et al.
(2016), where both studies had more complete VOC measurements). Finally, the
cycloalkenes/alkenes originate from numerous anthropogenic sources (e.g., McDonald et al.,
2018; Simpson et al., 2020). One subclass of cycloalkenes includes monoterpenes. Similar to the
comparison of isoprene between PTR-MS and WAS, the difference in monoterpenes between
these two measurements increases with increasing $NO_x$ (Figure S6). As the interfering
compound(s) measured by the PTR-MS and whether they are oxygenated or not is not known,
only the WAS monoterpenes are used in this analysis of calculating R(VOC). Assuming the
limonene rate constant, the difference between the PTR-MS and WAS monoterpenes raises the
terpene reactivity by $0.05 – 0.30$ s$^{-1}$. Though this does not include any associated photochemical
products from the oxidation of monoterpenes and can improve the closure, it does not explain the
total missing reactivity ($1.4 – 2.1$ s$^{-1}$). Thus, the missing R(VOC) is most likely a combination of
OVOCs and cycloalkenes/alkenes.

**3.4 Sources of PNs over SMA**

As shown in Figure 2, ΣPNs account for a larger fraction of the total $NO_z$ budget than

ΣANs. ΣPNs are known to be a temporary sink of $NO_x$ and radicals (R(O)O$_2$·) due to their short
thermal lifetime (~1 hr). Thus, the $NO_x$ emitted in SMA is being transported regionally, impacting
the P(O$_x$).





In Figure 3b, ΣPNs shows some correlation with formaldehyde. Both are secondary
products from the photooxidation of VOCs and have short lifetimes, leading to the correlation.
However, above 4 ppbv formaldehyde, the correlation shifts as ΣPNs increases more rapidly than
formaldehyde. As shown in Figure S7, this change in the relationship between ΣPNs versus
formaldehyde is due to changes in the competition in the reaction of the acyl peroxy radical
$(R(O)O_2^{\cdot})$ between $NO_2$ and NO. At low NO-to-$NO_2$ ratios, R8 is more favorable, leading to more
efficient production of PNs over formaldehyde. As NO-to-$NO_2$ ratios increase (NO becomes
comparable to $NO_2$), R9 becomes more dominant, leading to less production of PNs.
To further explore the sources of both PAN and the higher ΣPNs, the F0AM model (Wolfe
et al., 2016) was used to predict ΣPNs, constrained by the observed VOCs precursors (Table 2).
F0AM shows minimal bias in the predicted formaldehyde, $NO_2$, and OH (Figure S8). As discussed
in Sect. 3.3, though, there is missing R(VOC) of $1.7^{+1.1}_{-0.4}s^{-1}$. A sensitivity analysis in adding this
missing reactivity to F0AM on predicted OH and formaldehyde was conducted (Sect. S4 and
Figure S9 – S10). Both OH and formaldehyde are found to be buffered with the addition of this
low amount of R(VOC). Thus, though there is good agreement in these intermediate products
between observation and F0AM, this analysis for the sources of PAN and higher ΣPNs is expected
to be a lower limit. This missing R(VOC) is further observed in the F0AM-predicted higher PNs
(ΣPNs-PAN) versus formaldehyde, as a general underestimation in the total higher PNs compared
to observations is observed (Figure 5a). PAN was excluded as F0AM overestimated the mixing
ratios of PAN by approximately a factor of 2 (Figure S8e). Note, F0AM also overpredicted the
PPN mixing ratios, but to a lesser extent than PAN (~50%; Figure S8f). The differences in
predicted versus observed PNs may be associated with assumed background, dilution, and/or
temperature used to reach steady-state (Schroeder et al., 2020). Thus, the results from F0AM will





provide qualitative insight into sources and chemistry that should be investigated to better
understand PN chemistry in SMA.

The classes of compounds producing higher PNs in F0AM are shown in Figure 5b. The

classes of compounds were selected from the parent VOC which was oxidized into the higher PN
(Table S2). Individual PNs with high contributions and/or are typically measured (PPN, PBzN,
and MPAN (methacryloyl peroxy nitrate)) or are a large fraction of PNs but have yet to be
measured in ambient conditions (PHAN) are shown without any connection to the parent VOC.
Further, both PHAN and PPN have numerous precursors while many of the other higher PNs
modeled by F0AM are generally associated with one precursor. At high $NO_x$ mixing ratios, the
more reactive VOCs (aromatics, terpenes) contribute a large fraction of the higher PNs (>35% for
$NO_x$ > 4 ppbv). As the air moves away from SMA (lower $NO_x$ mixing ratios), contributions of
higher PNs from longer-lived compounds (e.g., alkanes) and later generation oxidation products
start dominating.

An interesting trend is observed for PPN and PHAN. Both peroxy acyl radicals for PPN

and PHAN ($C_2H_5C(O)O_2^{\cdot}$ and $CH_2(OH)C(O)O_2^{\cdot}$, respectively) are products from photooxidation
of many VOCs, including aromatics, alkanes, and methyl ethyl ketone (MEK). However, the
fractional contribution of PPN to higher PNs remains constant with decreasing $NO_x$ while the
fractional contribution of PHAN increases with decreasing $NO_x$ (Figure 5b). This stems from the
sources of $C_2H_5C(O)O_2^{\cdot}$ versus $CH_2(OH)C(O)O_2^{\cdot}$. The MCM mechanism, which is used for
F0AM, produces $C_2H_5C(O)O_2^{\cdot}$ from the photooxidation from both short- and long-lived species
(isoprene, C8-aromatics, toluene, ethanol, MEK, propane, and C4-alkanes) while
$CH_2(OH)C(O)O_2^{\cdot}$ is produced from the photooxidation of isoprene and ethene. For
$CH_2(OH)C(O)O_2^{\cdot}$, the production is through minor channels in the photooxidation of isoprene



(~3% yield directly from isoprene and ~20% as a secondary product (Galloway et al., 2011)).
Ethene is relatively long-lived, with a lifetime ~7 hrs (OH = $5\times10^6$ molec. cm$^{-3}$) leading to the
delay in the production of PHAN.
The results here in general indicate more speciated measurements of higher PNs are
needed. However, as highlighted in Figure 5, improved detection of or measurements of PBzN,
PHAN, and MPAN would allow for furthering our knowledge in PNs chemistry in urban
environments and their role in controlling $O_x$ production.
A qualitative investigation of the precursors of PAN predicted by F0AM are shown in
Figure 5c. This provides a basis for further investigation of the sources over the SMA region for
PAN as (a) F0AM over-predicts PAN, as noted above, (b) ethanol is currently estimated, similar
to Schroeder et al. (2020), and (c) R(VOC) in F0AM is low due to missing precursors. Like the
higher PNs, highly reactive R(VOC) contributes a large portion of the PAN budget at high $NO_x$.
The short-lived compounds contribute ~80% of PAN over SMA at the highest $NO_x$ mixing ratios.
At lower $NO_x$ mixing ratios, moving away from SMA, longer-lived compounds, such as ethanol,
contribute the most towards PAN production (~70%).
One of the interesting contributions not typically observed for PAN is MEK, which also
contributes to PPN and PHAN. In prior studies, MEK mixing ratios were typically 0.5 to 2.0 ppbv
(Bon et al., 2011; de Gouw et al., 2018; Liu et al., 2015). Over the SMA, 1.5 ppbv of MEK was
observed on average with values as high as 8.3 ppbv. Due to the long lifetime of MEK (~30 hrs
for the average photolysis rate measured and OH = $5\times10^6$ molec. cm$^{-3}$), the high mixing ratios of
MEK are most likely due to direct emissions (e.g., de Gouw et al., 2005; Liu et al., 2015). Thus,
there are potentially large sources of MEK in SMA that need to be considered in properly
representing PAN chemistry.





Another potentially important compound for PAN production is ethanol. However, this
compound was not measured during KORUS-AQ; instead, it was estimated based on previous
ground-based observations, similar to Schroeder et al. (2020). Ethanol is considered to mainly
come from vehicle emissions (e.g., Millet et al., 2012) and potentially cleaning agents (e.g.,
McDonald et al., 2018). As ethanol use is predicted to increase in the future (e.g., de Gouw et al.,
2012) and cleaning agents and other volatile chemical products appear to scale with population
(Gkatzelis et al., 2021), ethanol and MEK may continue contributing towards the PAN budget in
the SMA in the future.
As a note, two other compounds potentially important for PAN production that were not
measured on the DC-8 during KORUS-AQ include methylglyoxal and biacetyl (LaFranchi et al.,
2009). In a forested environment that was partially impacted by urban outflow, these two
components contributed on average 25% of the PAN budget (LaFranchi et al., 2009). In urban
environments, methylglyoxal is believed to mainly originate from aromatic oxidation (Ling et al.,
2020); whereas, biacetyl is believed to come from anthropogenic emissions (Xu et al., 2023).
Further, as discussed in Sect. 4.3, these two compounds may potentially be important missing $HO_x$
sources, as well. Thus, measurements of these two compounds along with ethanol is necessary to
better understand PAN chemistry.

**4. Observational constraints of the $HO_x$ and $O_x$ budget over SMA**

As highlighted in Figure S1, the three factors impacting instantaneous $P(O_x)$ are R(VOC),
$P(HO_x)$, and $NO_x$ loss processes. In Sect. 3, the $NO_x$ loss processes were investigated and provided
a constraint for R(VOC) to improve the investigation of $P(O_x)$. With R(VOC) constrained, the
$RO_2^.$ concentration can be estimated, providing a means to calculate the net $P(O_x)$ and to
investigate the major reactions leading to $O_x$ loss and total $HO_x$ ($OH + HO_2 + RO_2^. + R(O)O_2^.$)
loss. With the latter, this allows for an investigation of the major $P(HO_x)$ reactions, assuming
$L(HO_x)$ equals $P(HO_x)$ (see Eq. 1 – 7 in Sect. 2.3).

## 560    4.1 Net $O_x$ production and sources of $O_x$ loss

Using the total $R(VOC)$ from Sect. 3.3 (Figure 4a), the net $P(O_x)$ (Eq. 1 – 2) over SMA

during KORUS-AQ has been determined (Figure 6a). The net $P(O_x)$ peaked at 10.3 ppbv $hr^{-1}$ at
~8 ppbv $NO_x$. If only the measured and estimated $R(VOC)$ from F0AM secondary products is used
to calculate net $P(O_x)$, the value decreases to 8.8 ppbv $hr^{-1}$, but at the same $NO_x$ mixing ratio. This
value is similar to values observed in other urban locations around the world (~2 – 20 ppbv $hr^{-1}$),
showing that many urban areas are still impacted by high $P(O_x)$ values (Brune et al., 2022; Griffith
et al., 2016; Ma et al., 2022; Ren et al., 2013; Schroeder et al., 2020; Whalley et al., 2016, 2018).

The $NO_x$ distribution over SMA (Figure 1) shows a large area (~127.53°E to 127.18°E, or

~39 km) is near the $NO_x$ mixing ratio with the maximum $P(O_x)$ (Figure 6). Thus, a large portion
of the SMA will have high instantaneous $P(O_x)$ of ~10 ppbv $hr^{-1}$. As the median wind speed over
SMA during KORUS-AQ was ~5 m $s^{-1}$, an air parcel would remain at the highest $P(O_x)$ for ~2 hrs,
leading to ~20 ppbv $O_3$ being produced (not including dilution). This agrees with the ~20 ppbv
increase in $O_3$ observed over the Taehwa Research Forest supersite between midday and afternoon
overpasses by the DC-8 during KORUS-AQ (Crawford et al., 2021). Thus, though there is a
substantial $O_3$ background observed over SMA (Colombi et al., 2023; Crawford et al., 2021), a
large contribution of the $O_3$ is due to photochemical production.

The major reactions leading to $O_x$ loss ($L(O_x)$) are shown in Figure 6b. The two major

reactions that lead to $O_x$ loss are net R8 (light and dark red), or the net production of PNs (which




includes losses), and R11, reaction of $NO_2$ with OH (blue) (see Table 1). Note, as discussed in
Sect. 2.2, for the budget analysis conducted here, PAN and PPN were constrained to observations.
At high $NO_x$ (near emissions, ~30 ppbv), R11 ($NO_2$ + OH) dominates the $L(O_x)$ budget (> 60%),
with net R8 (net PAN, dark red, and higher PNs, light red) contributing ~25%, and R12 – 14
accounting for the remaining 15% of $O_x$ loss. As $NO_x$ mixing ratios decrease (moving away from
emissions), the net R8 reaction, producing both PAN and higher PNs, starts contributing to larger
total $L(O_x)$, ranging from 30 – 40%. Furthermore, the net R8 reaction contribution towards $L(O_x)$
remains relatively constants with $NO_x$ mixing ratios as the contribution from R11 (OH + $NO_2$)
decreases. At $NO_x$ mixing ratios < 3 ppbv is when non-$NO_x$ reactions (R12 – 14) contribute greater
than 30% of the $L(O_x)$ budget. Thus, proper representation of PAN and higher PNs, both in
precursors and speciation, is important in properly understanding the $O_x$ budget in SMA.

**4.2 $HO_x$ loss over the SMA**

Similar to $L(O_x)$, the major reactions leading to $L(HO_x)$ over the SMA during KORUS-AQ

were the reactions of $NO_x$ with $HO_x$, specifically $NO_2$ with OH (R11) and net PAN (dark red) and
higher PNs (light red) production (R8). Reaction R11 is most important for $NO_x$ mixing ratios
greater than 15 ppbv (50 – 65%). Between 5 and 15 ppbv, R11 is comparable to the net PN
production (R8), where R11 comprises 35 – 50% of $L(HO_x)$ while net R8 (sum of higher $\sum$PNs
and PAN) comprises 30 – 40% of $L(HO_x)$. At lower $NO_x$ mixing ratios, R11 is always smaller for
$L(HO_x)$ than net R8, where R11 is about a factor of 2 lower than net R8. Production of $\Sigma$ANs
played a minor role due to the low $\alpha_{eff}$.

The self-reaction of $HO_x$ species (R15 – R16) contributes minimally to $L(HO_x)$ (less than

10%) for $NO_x$ mixing ratios greater than 8 ppbv. At lower $NO_x$ mixing ratios, R16 starts



dominating $L(HO_x)$ budget, increasing from 8% at 8 ppbv to 50% of $L(HO_x)$ at $NO_x$ mixing ratios
less than 2 ppbv. Reaction R15 remains relatively small for the $L(HO_x)$ budget, only reaching 7%
of the $L(HO_x)$ budget at $NO_x$ mixing ratios less than 2 ppbv.

**4.3 Sources of $HO_x$ over SMA**

The analysis conducted leads to the ability to constrain $HO_x$ losses over the SMA during

KORUS-AQ. This is important as not all typical $HO_x$ sources were measured on the DC-8 during
the project (e.g., nitrous acid, or HONO), and $HO_x$ production rates directly impacts $P(O_x)$ (Figure
S1). Prior studies (e.g., Griffith et al., 2016; Tan et al., 2019; Whalley et al., 2018) have
demonstrated that in urban environments, sources of $HO_x$ include photolysis of $O_3$ and subsequent
reaction with water vapor, formaldehyde photolysis, and HONO photolysis. Furthermore, recent
studies have highlighted the potential importance of typically non-measured OVOCs in their
contribution to $P(HO_x)$ and subsequent $P(O_x)$ in an urban environment (Wang et al., 2022). To
constrain the $P(HO_x)$ over SMA during KORUS-AQ, the $P(HO_x)$ was assumed to be equal to the
observationally constrained $L(HO_x)$. Then, $P(HO_x)$ was calculated for the measurements on the
DC-8, including photolysis of $O_3$, formaldehyde, $H_2O_2$, and other measured OVOCs (Table 2).

Comparing the calculated $P(HO_x)$ and $L(HO_x)$, ~1.5 ppbv hr$^{-1}$ $P(HO_x)$ (range 1.3 – 1.8 ppbv

hr$^{-1}$) is not accounted for, leading to ~45% of the necessary $L(HO_x)$ to maintain steady-state
(Figure 7). For the calculated $P(HO_x)$ budget, $O_3$ and formaldehyde photolysis contributed ~50%
and 40% of the budget, respectively, with the remainder coming from photolysis of $H_2O_2$ and other
measured OVOCs. Accounting for the unobserved $P(HO_x)$, $O_3$ and formaldehyde photolysis
contributed ~25% and ~20%, respectively.



Potential missing sources of $P(HO_x)$ are briefly speculated here. First, one potential source
is the photolysis of methylglyoxal. Using the F0AM predicted methylglyoxal, as it was not
measured on the DC-8, methylglyoxal would contribute ~0.24 ppbv hr$^{-1}$ $P(HO_x)$, or ~16% of the
unobserved $P(HO_x)$. Another OVOC not measured on the DC-8 and expected to originate from
anthropogenic emissions and not from chemistry is 2,3-butanedione, or biacetyl (de Gouw et al.,
2018; Grosjean et al., 2002; Schauer et al., 2002; Xu et al., 2023; Zhou et al., 2020). Prior studies
observed 20 – 400 pptv of biacetyl (de Gouw et al., 2018; Xu et al., 2023), correspond to 0.04 –
0.74 ppbv hr$^{-1}$, or 3 – 49% of the unobserved $P(HO_x)$. Thus, between these two OVOCs, 19 – 66%
of the unobserved $P(HO_x)$ could be explained. Other unmeasured OVOCs could potentially
contribute to the observed $P(HO_x)$ (e.g., Wang et al., 2022); however, there is less constraints both
on the speciation and photolysis rates for these OVOCs (e.g., Mellouki et al., 2015). Finally,
HONO could contribute to this observed $P(HO_x)$. Up to 700 pptv of HONO was observed in SMA
during KORUS-AQ (Gil et al., 2021), though, this would quickly photolyze to the altitudes the
DC-8 flew over SMA (Tuite et al., 2021). Even at 50 – 100 pptv HONO, photolysis of HONO
would lead to 0.2 – 0.4 ppbv hr$^{-1}$ $P(HO_x)$, or 13 – 27% of the unobserved $P(HO_x)$. Thus, between
methylglyoxal, biacetyl, and HONO, between 32 – 92% of the unobserved $P(HO_x)$ could be
accounted for. This analysis highlights the importance of measuring these $HO_x$ sources to better
understand and constrain $O_x$ chemistry in SMA and other urban environments.
One note about this analysis is that particulate matter collected onto the downwelling CAFS
optics during KORUS-AQ (see Sect. S5, Table S3, and Figure S11). Corrections of up to 20%
were determined, and the associated uncertainties were also increased by 20% due to the
corrections. Thus, the exact amount of unmeasured $P(HO_x)$ is potentially smaller than discussed.



**5. Conclusions and Implications**

In the Seoul Metropolitan Area (SMA), the ozone ($O_3$) mixing ratio often exceeds current standards and is increasing. Many processes can impact the $O_3$ mixing ratios and exceedances. Here, the processes that impact instantaneous $O_3$ production ($P(O_x)$, where $O_x$ is $O_3 + NO_2$ to account for possible $O_3$ titration) were investigated for observations collected on the NASA DC-8 during the 2016 NIER/NASA Korea United-States Air Quality (KORUS-AQ) study. The observations indicate missing oxidized $NO_x$ products ($NO_z$) that include both the short-lived peroxy nitrates ($\Sigma$PNs) and alkyl and multi-functional nitrates ($\Sigma$ANs). $\Sigma$PNs contributed the most for the organic $NO_z$ species. Only ~50% of the $\Sigma$PNs were speciated over SMA, which is atypical as prior studies typically show closure between the speciated and total PN measurements.

The un-speciated $\Sigma$PNs and $\Sigma$ANs were used to constrain the missing volatile organic compound (VOC) reactivity (R(VOC)), as R(VOC) is important in constraining the instantaneous $P(O_3)$. The missing R(VOC) was found to be 1.4 to 2.1 $s^{-1}$. The F0AM box model further supports the role of unmeasured $\Sigma$PNs as an important temporary $NO_x$ and radical sink over SMA. F0AM predicts ~50% of the higher $\Sigma$PNs (higher $\Sigma$PNs = $\Sigma$PNs – PAN), indicating missing R(VOCs) may explain the other 50%. Constraints from both the $\Sigma$PNs and $\Sigma$ANs suggest that this missing R(VOC) would include oxygenated VOCs (OVOCs), including aldehydes such as octanal and nonanal from cooking, and alkenes from anthropogenic emissions.

With the constraints on the R(VOC), the net instantaneous $P(O_x)$ was determined for SMA. It was found to peak at ~10 ppbv $hr^{-1}$ at ~8 ppbv $NO_x$. A large fraction of the SMA area was, on average, at this mixing ratio of $NO_x$, indicating high local $P(O_x)$. This supports the increase of ~20 ppbv of $O_3$ observed in a downwind site (Taehwa Research Forest supersite) from midday to afternoon during KORUS-AQ.



With the comprehensive measurements on-board the DC-8, the F0AM model results, and
the observationally constrained R(VOC), a budget analysis on the sinks of $O_3$ ($L(O_x)$) and $HO_x$
($L(HO_x)$, where $HO_x = OH + HO_2 + RO_2^. + R(O)O_2^.$) was performed. Due to the high R(VOC),
type of VOC, and the $NO_2$-to-NO ratio, net ΣPNs production is surprisingly a large and important
sink of $O_x$ and $HO_x$ over SMA (~25 – 40% and 15 – 40% for $L(O_x)$ and $L(HO_x)$, respectively),
with production of $HNO_3$ and radical self-reactions accounting for the other $L(O_x)$ and $L(HO_x)$
losses. Net ΣPNs production as an important $L(O_x)$ and $L(HO_x)$ term is significant, as ΣPNs is a
temporary reservoir of both $NO_2$ and $R(O)O_2^.$ but has not traditionally been included in these
calculations. Downwind locations separated from the local $NO_x$ and VOC emissions of the SMA
will experience increased $P(O_x)$ due to the release of $NO_2$ and $R(O)O_2^.$. With the constraint of
$L(HO_x)$, $P(HO_x)$ was investigated, assuming steady-state, and unmeasured HONO plus
unmeasured OVOCs were found to be necessary to explain the missing $HO_x$ sources. Both sources
of $HO_x$ are either missing or highly uncertain in chemical transport models.
Though the high regional background and foreign sources of $O_3$ and its precursors elevate
the $O_3$ levels in SMA and potentially already causes the SMA to be in exceedance for $O_3$
concentrations, this study highlights the importance local, in-situ $P(O_x)$ to the SMA area, which
can further exacerbate the $O_3$ concentrations for SMA and the surrounding region. The results
support the observations of increasing $O_3$ with decreasing $NO_x$ that has been observed for SMA in
prior studies. Further, the study highlights the important role of unmeasured VOCs and OVOCs
and the necessity to understand their sources and role in $NO_x$ and $O_3$ chemistry. Further, the study
demonstrates the interplay of direct emissions or secondary production of PN precursors and its
role in net $P(O_x)$. Attempts at specifically reducing the sources of PN may adversely impact net
$P(O_x)$, as lower net PN chemistry may increase $O_3$ due to more $NO_2$ being available.



**Competing Interests**

At least one of the (co-)authors is a member of the editorial board of Atmospheric Chemistry and Physics.

**Acknowledgements**

The authors acknowledge Michelle Kim, Alex Teng, John Crounse, and Paul O. Wennberg for their measurements with CIT-CIMS (HNO$_3$, multifunctional alkyl nitrates, and OVOCs), William H. Brune for his measurements with ATHOS (OH, OH reactivity), Alan Fried for his measurements with CAMS (CH$_2$O and C$_2$H$_6$), Paul Romer-Present for his contribution to collecting data with TD-LIF, Sally Pusede for her contributions to collecting data with DACOM and DLH, and Andrew J. Weinheimer for his measurements of NO, O$_3$, and NO$_y$. The PTR-MS instrument team (P. Eichler, L. Kaser, T. Mikoviny, M. Müller) are acknowledged for their support.

**Funding**

BAN and KRT acknowledge NASA grant 80NSSC22K0283. LGH and YL acknowledge NASA grant NNX15AT90G for the PAN measurements. SRH and KU were supported by NASA grant NNX15AT99G for photolysis measurements. AW acknowledges support by the Austrian Federal Ministry for Transport, Innovation, and Technology (bmvit-FFG-ASA) for the PTR-MAS measurements. PCJ and JLJ were supported by NASA 80NSSC21K1451 and 80NSSC23K0828.

**Data Availability**





Version R6 1-min merged data used in this analysis available at
DOI:10.5067/Suborbital/KORUSAQ/DATA01. The F0AM setup file, input file, and output files
are all available at https://doi.org/10.5281/zenodo.10723227.

**Author Contribution**
BAN, KRT, and JHC designed the experiment and wrote the paper. BAN and KRT analyzed the
data. KRT ran the F0AM model and KRT and BAN analyzed the model output. BAN, DRB, PCJ,
RCC, JPD, GSD, SRH, LGH, JLJ, K-EK, YL, IJS, KU, and AW collected and QA/QC the data
during KORUS-AQ. All authors contributed to the writing and editing of the paper.



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



**Tables**
**Table 1.** Reactions described in text along with associated rate constants and references for those
rate constants.

| | *Reaction* | *Reaction Rate* | *Reference* |
|---|---|---|---|
| R1a | $VOC + OH \xrightarrow{O_2} RO_2^{\cdot}$ | Varies | Atkinson (2003); Atkinson and Arey(2003); Atkinson et al. (2006); Bohn and Zetzsch (2012); Sprengnether et al. (2009) |
| R1b | $VOC + h\nu \xrightarrow{O_2} RO_2^{\cdot}$ | Varies/Measured | Shetter & Müller (1999) |
| R2a | $RO_2^{\cdot} + NO \rightarrow (1-\alpha)\ RO^{\cdot} + (1-\alpha)\ NO_2$ | $2.7 \times 10^{-11} \times \exp(390/T)$ | Burkholder et al. (2020) |
| R2b | $RO_2^{\cdot} + NO \rightarrow \alpha\ RONO_2$ | $2.7 \times 10^{-11} \times \exp(390/T)$ | Burkholder et al. (2020) |
| R3 | $NO_2 + h\nu \rightarrow NO + O(^3P)$ | Measured on DC-8 | Shetter & Müller (1999) |
| R4 | $O(^3P) + O_2 \rightarrow O_3$ | $3.2 \times 10^{-11} \times \exp(67/T)$ | Saunders et al. (2003) |
| R5 | $RO^{\cdot} + O_2 \rightarrow R(O) + HO_2$ | Assumed Instantaneous | |
| R6 | $HO_2 + NO \rightarrow OH + NO_2$ | $3.45 \times 10^{-12} \times \exp(270/T)$ | Saunders et al. (2003) |
| R7 | $RCHO + OH \xrightarrow{O_2} R(O)O_2^{\cdot}$ | Varies | Atkinson (2003); Atkinson and Arey(2003); Atkinson et al. (2006) |
| R8[a] | $R(O)O_2^{\cdot} + NO_2 \leftrightarrow R(O)O_2NO_2$ | F: $8.69 \times 10^{-12}$ cm$^3$ molec.$^{-1}$ s$^{-1}$ R: $4.30 \times 10^{-4}$ s$^{-1}$ | Burkholder et al. (2020) |
| R9 | $R(O)O_2^{\cdot} + NO \rightarrow RO_2^{\cdot} + NO_2$ | $8.1 \times 10^{-12} \times \exp(270/T)$ | Burkholder et al. (2020) |
| R10 | $O_3 + NO \rightarrow O_2 + NO_2$ | $2.07 \times 10^{-12} \times (-1400/T)$ | Burkholder et al. (2020) |
| R11[b] | $OH + NO_2 \rightarrow HNO_3$ | $1.24 \times 10^{-11}$ cm$^3$ molec.$^{-1}$ s$^{-1}$ | Burkholder et al. (2020) |
| R12 | $O_3 + h\nu \xrightarrow{H_2O} 2O(^1D)$ | hv measured on DC-8; $2.14 \times 10^{-10}$ cm$^3$ molec.$^{-1}$ s$^{-1}$ | Shetter & Müller (1999); Saunders et al. (2003) |
| R13 | $O_3 + OH \rightarrow HO_2 + O_2$ | $1.7 \times 10^{-12} \times \exp(-940/T)$ | Saunders et al. (2003) |
| R14 | $O_3 + HO_2 \rightarrow OH + 2O_2$ | $1.0 \times 10^{-14} \times \exp(-490/T)$ | Burkholder et al. (2020) |
| R15[b] | $HO_2 + HO_2 \xrightarrow{H_2O} H_2O_2$ | $5.06 \times 10^{-12}$ cm$^3$ molec.$^{-1}$ s$^{-1}$ | Saunders et al. (2003) |
| R16 | $HO_2 + RO_2 \rightarrow Products$ | $2.91 \times 10^{-13} \times \exp(1300/T)$ | Saunders et al. (2003) |
| R17 | $HO_2 + OH \rightarrow Products$ | $4.80 \times 10^{-11} \times \exp(250/T)$ | Burkholder et al. (2020) |



| R18[b] | OH+NO → HONO | $7.40\times10^{-12}$ cm³ molec.⁻¹ s⁻¹ | Burkholder et al. (2020) |
|---|---|---|---|
| R19 | HO₂+R(O)O₂ → Products | $4.30\times10^{-13}\times\exp(1040/T)$ | Burkholder et al. (2020) |

[a]Only showing forward (F) and reverse (R) rate constant at 298 K and 1013 hPa and being a
termolecular reaction.
[b]Termolecular reaction; only showing rate at 298 K and 1013 hPa



**Table 2.** List of instruments, compounds measured, accuracy/precision, and associated references
used in this study.

| Instrument | Species | References |
|---|---|---|
| University of California, Irvine, Whole Air Sampler (WAS) | Ethane, Ethene, Ethyne, Propane, Propene, i-Butane, n-Butane, 1-Butene, i-Butene, trans-2-Butene, cis-2-Butene i-Pentane, n-Pentane, 1,3-Butadiene, Isoprene, n-Hexane, n-Heptane, n-Octane, n-Nonane, n-Decane, 2,3-Dimethylbutane, 2-Methylpentane, 3-Methylpentane, Cyclopentane, Methylcyclopentane, Cyclohexane, Methylcyclohexane, Benzene, Toluene, m+p-Xylene, o-Xylene, Ethylbenzene, Styrene, i-Propylbenzene, n-Propylbenzene, 3-Ethyltoluene, 4-Ethyltoluene, 2-Ethyltoluene, 1,3,5-Trimethylbenzene, 1,2,4-Trimethylbenzene, 1,2,3-Trimethylbenzene, α-Pinene, β-Pinene, Methyl nitrate, Ethyl nitrate, i-Propyl nitrate, n-Propyl nitrate, 2-Butyl nitrate, 3-Pentyl nitrate, 2-Pentyl nitrate, 3-Methyl-2-Butyl nitrate | Simpson et al. (2020) |
| The Pennsylvania State University Airborne Tropospheric Hydrogen Oxides Sensor (ATHOS) | OH, $HO_2$, OH Reactivity | Faloona et al. (2004), Mao et al. (2009), Brune et al. (2019) |
| University of California, Berkeley, Thermal Dissociation-Laser Induced Fluorescence (TD-LIF) | $NO_2$, ΣPNs, ΣANs | Thornton et al. (2000), Day et al. (2002), Wooldridge et al. (2010) |
| NASA Langley Diode Laser Hygrometer (DLH) | $H_2O$ | Diskin et al. (2002) |
| NASA Langley Diode Laser Spectrometer Measurements (DACOM) | CO, $CH_4$ | Sachse et al. (1987) |
| University of Colorado, Boulder, Compact Atmospheric Multi-species Spectrometer (CAMS) | $CH_2O$, $C_2H_6$ | Richter et al. (2015), Fried et al. (2020) |
| Gwangju Institute of Science and Technology Korean Airborne Cavity Enhances Spectrometer (K-ACES) | CHOCHO | Min et al. (2016), D. Kim et al. (2022) |



| NCAR CCD Actinic Flux Spectroradiometers (CAFS) | j-values | Shetter & Müller (1999) |
|---|---|---|
| Georgia Institute of Technology Chemical Ionization Mass Spectrometer (GT) | $SO_2$, PAN, PPN, APAN, PBzN | Kim et al. (2007), Lee et al. (2022) |
| University of Colorado, Boulder, High-Resolution Time-of-Flight Aerosol Mass Spectrometer | $pNO_3$ | Nault et al. (2018), Day et al. (2022) |
| NCAR 4-Channel Chemiluminescence Instrument (NCAR) | NO, $NO_2$, $O_3$, $NO_y$ | Weinheimer et al. (1994) |
| California Institute of Technology Chemical Ionization Mass Spectrometer (CIT) | Butene Hydroxynitrates, Butadiene Hydroxnitrates, Ethene Hydroxynitrates, Ethanal Nitrate, Isoprene Hydroxynitrates, Propene Hydroxynitrates, Propanal Nitrate, $CH_3OOH$, Peroxyacetic Acid, $HNO_3$, Hydroxyacetone, $H_2O_2$ | Crounse et al. (2006), Teng et al. (2015) |
| University of Oslo Proton Transfer Reaction Time-of-Flight Mass Spectrometer (PTR-MS) | Methanol, Acetaldehyde, Acetone+Propanal, Isoprene, MVK+MACR+ISOPOOH, Benzene, Toluene, C8-alkylbenzenes, Monoterpenes, MEK | Müller et al. (2014) |
| NSRC Meteorological and Geographical Data | Latitude, Longitude, Altitude, Temperature, Pressure | Crawford et al. (2021) |


**Figures**

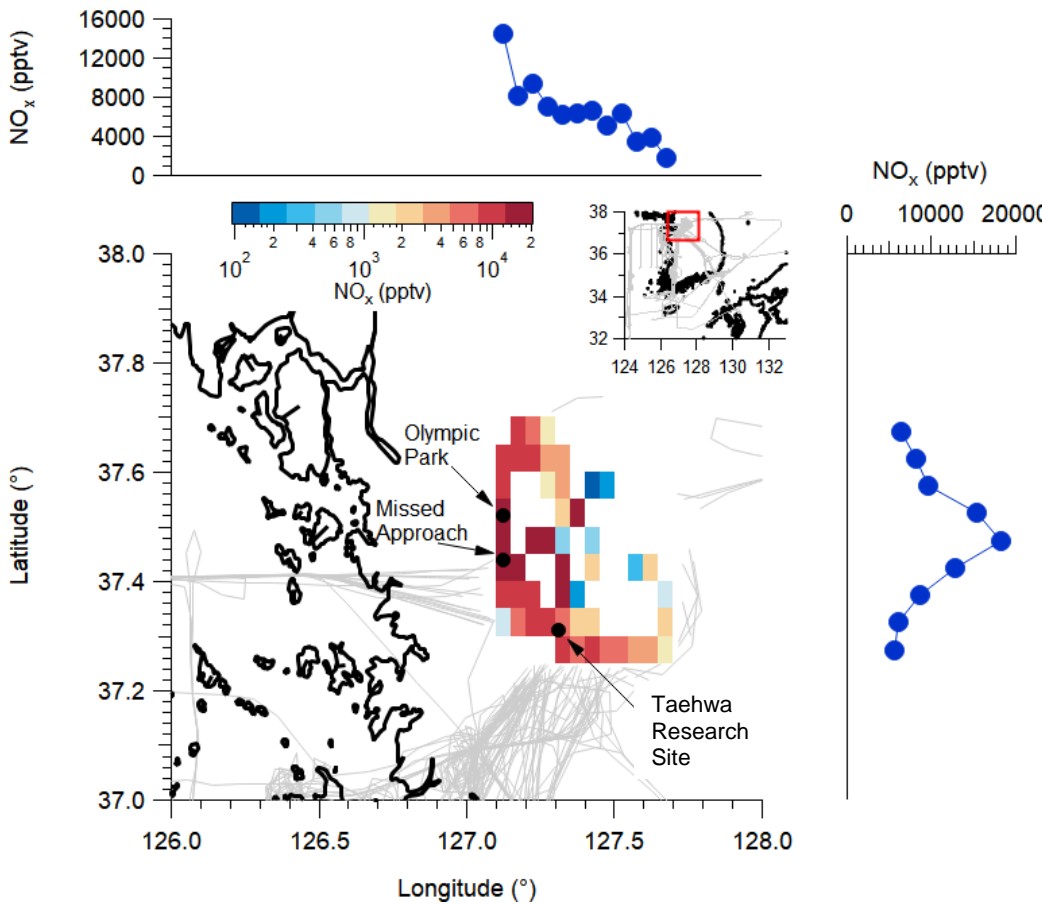

**Figure 1.** Binned $NO_x$ mixing ratios observed on the NASA DC-8 during the KORUS-AQ campaign. Note, the color bar scale is logarithmic. The binning is along the flight paths of the NASA DC-8 for any observations collected below 2.0 km and after 11:00 local time. The rest of the NASA DC-8 flight paths not included in the analysis are shown in grey. Three key areas from KORUS-AQ are highlighted—the Olympic Park ground site, the airfield where the NASA DC-8 conducted routine missed approaches, and the Taehwa Research ground site. The histograms above and to the left are the distribution of $NO_x$ mixing ratios longitudinally and latitudinally, respectively.

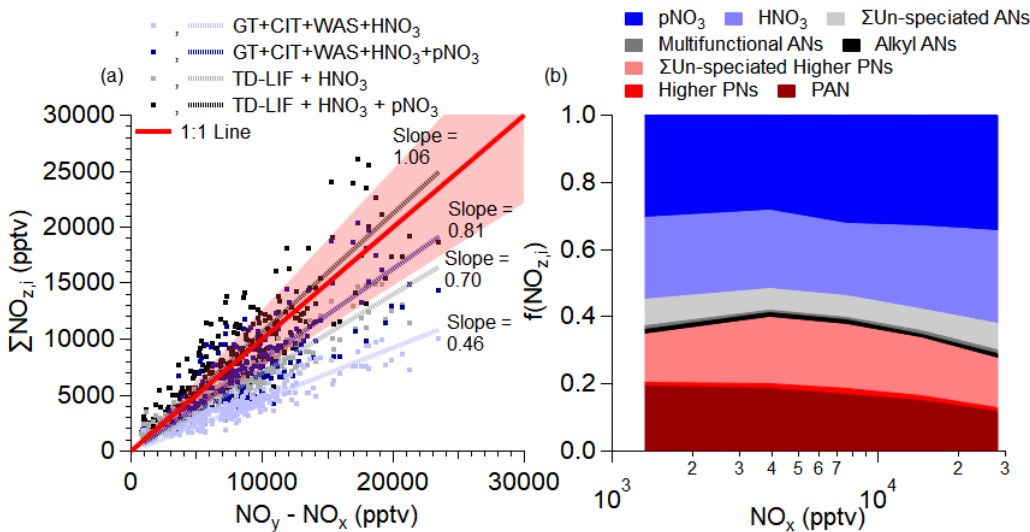

**Figure 2.** (a) Scatter plot of the summation of individual $NO_z$ ($NO_z$ is higher oxide $NO_x$ products) measured by GT, CIT, WAS, TD-LIF, and AMS versus $NO_z$ measured by difference between $NO_y$ and $NO_x$ (see Table 2 for compounds measured by each instrument). $NO_x$ is NO measured by NCAR and $NO_2$ measured by LIF. The observations are for when the DC-8 was over the SMA. (b) Average contribution of measured speciated $NO_z$ over the SMA during KORUS-AQ versus $NO_x$. Higher PNs is PPN + APAN + PBZN. ΣUn-speciated PNs is total peroxnitrates from TD-LIF minus total measurement from GT. Alkyl $RONO_2$ is the total small alkyl nitrate measurements from WAS. Multifunctional $RONO_2$ is the total measurements from CIT. ΣUn-speciated ANs is the total alkyl nitrates from TD-LIF minus total $RONO_2$ from CIT and WAS.

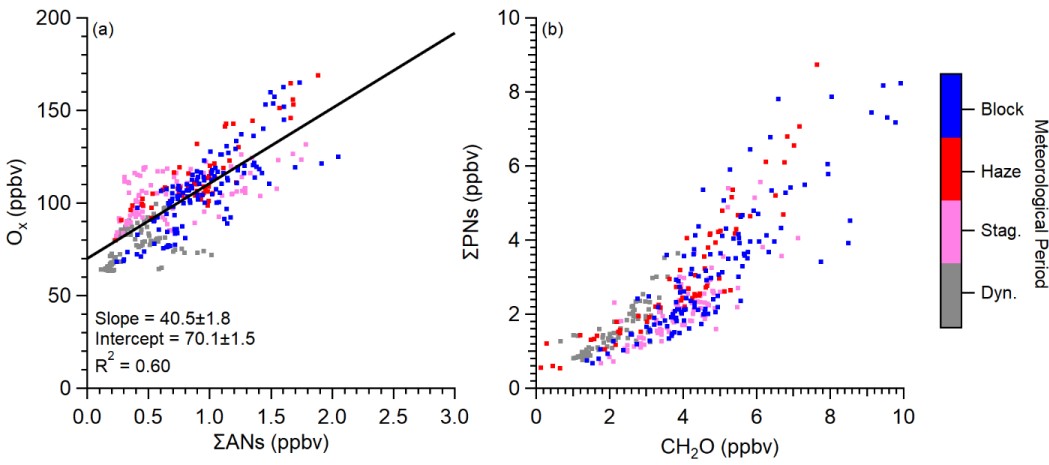

1212

**Figure 3.** Scatter plot of (a) $O_x$ versus $\Sigma$ANs and (c) $\Sigma$PNs versus formaldehyde ($CH_2O$) over SMA (see Figure 1 for area studied). Data is colored by meteorological periods discussed in Peterson et al. (2019). Data plotted here is after 11:00 am LT to minimize impact of growing boundary layer and nocturnal residual layer mixing. The curvature in (c) is further explored in Figure S7.

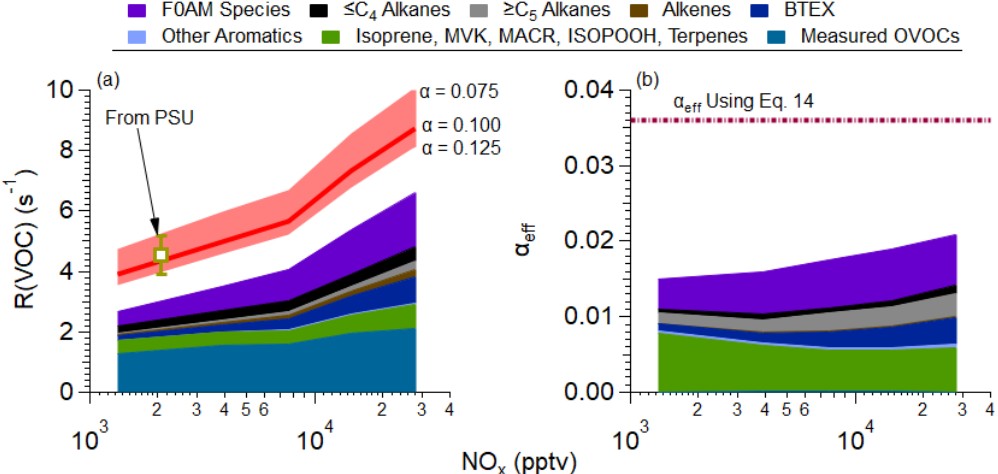

1218

**Figure 4.** (a) Binned VOC reactivity versus NO$_x$ observed over SMA during KORUS-AQ (see Figure 1 for the area studied). The measured observed R(VOC), labeled as "From PSU", where PSU is Pennsylvania State University, is the VOC reactivity calculated from the measured total OH reactivity with inorganic OH reactivity removed. As discussed in Brune et al. (2022), the OH reactivity has interferences at high NO$_x$ mixing ratios. The error bar is the uncertainty in the OH reactivity measurement (Brune et al., 2022). The red line represents the calculated unmeasured R(VOC), using Eq. 11, with an assumed $\alpha = 0.10$. The shaded area represents different calculated unmeasured R(VOC), assuming different $\alpha$ for the unmeasured R(VOC) (see Eq. 11). (b) The calculated effective $\alpha$ from observations versus NO$_x$. The dashed purple line is the effective $\alpha$ estimated from Eq. 10, using the slope from Figure 3a. For both (a) and (b), the colored stacked data is the calculated VOC reactivity (a) and weighted effective $\alpha$ (b). The values from (b) are calculated using Eq. 11. Finally, for both (a) and (b), F0AM species is the reactivity for compounds not measured on the DC-8 predicted by F0AM with an estimated $\alpha = 0.05$. The associated uncertainty in using different $\alpha$ for the F0AM predicted reactivity is explored in Figure S4.

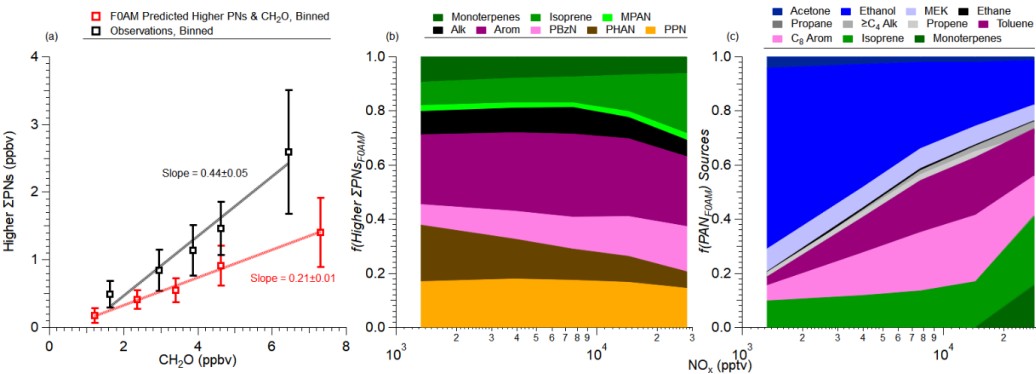

1233

**Figure 5.** (a) Scatter plot of binned higher ΣPNs calculated using F0AM (red) or binned higher
ΣPNs from observations (black) versus formaldehyde ($CH_2O$). Slopes shown are ODR fits to the
binned data. (b) Fractional contribution of the higher PNs predicted from F0AM versus $NO_x$. (c)
Fractional contribution of different precursors to PAN, predicted by F0AM versus $NO_x$. For both
(b) and (c), Alk is all alkanes, Arom is all aromatics, and ≥$C_4$ Alk is all alkanes with 4 or more
carbons. See Figure S8 for comparison of F0AM.

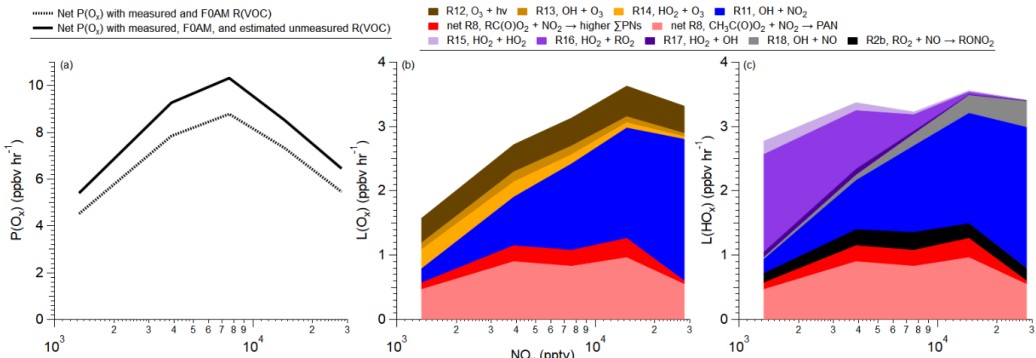

1240

**Figure 6.** (a) Net $O_x$ ($O_3$ + $NO_2$) production (see Eq. 1 and 2) predicted for SMA using measured and F0AM R(VOC) (dashed) or total R(VOC) (solid), from Figure 4a, versus $NO_x$. (b) Contribution of different reactions to the total $O_x$ loss versus $NO_x$. (c) Contribution of different reactions to total $HO_x$ ($HO_x$ = OH + $HO_2$ + $RO_2$ + R(O)$O_2$) loss versus $NO_x$. The predicted $RO_2$ comes from the total VOC reactivity calculated in Figure 4a assuming steady-state (Eq. 7), and $HO_2$ the acyl peroxy radicals are from F0AM results. Note for both (b) and (c), net RC(O)$O_2$ + $NO_2$ and net $CH_3C(O)O_2$ + $NO_2$ are described in Eq. 3. Radical reactions contributing < 1% to the $L(O_x)$ or $L(HO_x)$ are not included.



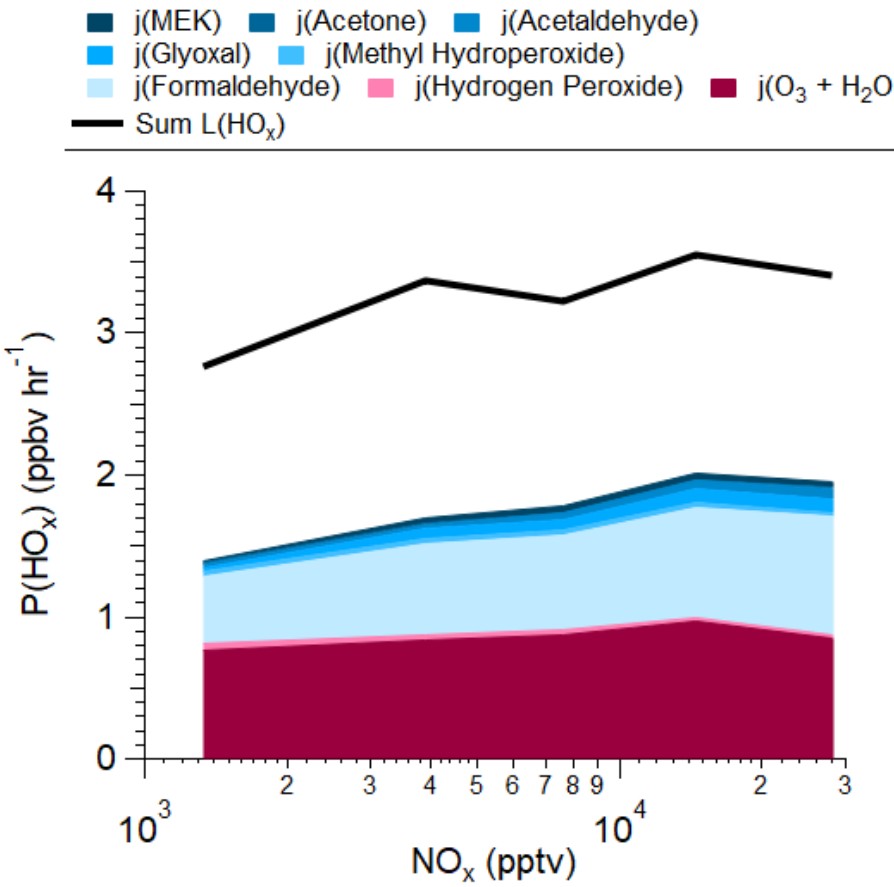

1249

**Figure 7.** Calculated HO$_x$ production from observations (colored stack) compared with the calculated HO$_x$ loss from Figure 6c over the SMA during KORUS-AQ.