# Peer review of "Using observed urban NOx sinks to constrain VOC reactivity and the ozone and radical"

_EGUsphere, 2024_

## Author Comment (AC1)

**Response to reviewers' comments on the paper "Using observed urban NOx sinks to constrain VOC reactivity and the ozone and radical budget in the Seoul Metropolitan Area"**

We thank the reviewers for their thorough and thoughtful reviews that have helped to improve and clarify our paper. For ease, comments from reviewers are in black, responses can be found in blue, and new text added to paper are in **bold blue**.

*Reviewer RC1 (Reviewer #2)*

1.0 Nault et al. describe $O_3$ production and its individual contributors in the Seoul Metropolitan Area based on airborne measurements with the NASA DC-8 aircraft during the KORUS-AQ campaign in 2016, as well as box model simulations using F0AM. The authors highlight three important aspects, which are the VOC reactivity, the production of $HO_x$ and the branching ratio of alkyl nitrates. A particular focus is put on the impact of unmeasured (O)VOCs, affecting underestimated peroxy and alkyl nitrates, and in turn deviations in $NO_x$ and radical sinks.

The paper is well written and interesting to read. I have some remaining questions and comments (see below). Once these are addressed, the paper would be a valuable contribution to literature and I recommend it for publication.

We thank the reviewer for the overall positive review and detailed comments. We have addressed each point below.

Major Comments

1.1 Does "unmeasured VOC" refer to species that are neither measured, nor represented in the model?

Unmeasured VOC is for compounds not represented in F0AM and not measured. We have defined this in the text at line 390 as:

**"Here, we are defining unmeasured R(VOC) as the reactivity not represented by measurements on the DC-8 or by F0AM predicted species and reactivity."**

1.2 Was Eq. (1) or Eq. (9) used to calculate $P(O_x)$ throughout the study? Could you present a comparison between the results of the different approaches?

We have better defined when we are using Eq. (1) vs Eq. (9), e.g., Eq. (9) is used to calculate the unmeasured reactivity (reactivity not measured by instrument nor predicted by F0AM) for Fig. 4 whereas we used Eq. (1) to calculate Net $P(O_x)$ in Fig. 6. We have updated both in text as well in caption when we are using each caption.

Also, we have added a SI section to compare the two different methods to calculate $P(O_x)$, starting at line 269 in the SI:

**"S6. Comparison in Calculating P(Ox)**

**Two different equations to calculate $P(O_x)$ are introduced in the main text – Eq. 1 and Eq. 9. Eq. 1 is more explicit as it is tracking the number of $O_x$ molecules formed from all reactions of $RO_2$ and $HO_2$ molecules with NO (and accounting for the fraction of reactions where $RO_2$ and NO form ANs); whereas, Eq. 9 is simplified version and takes the reactivity averaged α and γ for the environment and folds $HO_2$ into the R(VOC). Comparing the $P(O_x)$ from the two equations is shown in Figure S13. Since Eq. 1 is more explicit, it is approximately 24% higher than Eq. 9, as Eq. 9 does not directly account for $RO_2$ concentrations and assumes the total amount of $HO_2$ molecules formed. Eq. 1 is more accurate as it is not assuming the total amount of $HO_2$ formed and thus used when directly calculating $P(O_x)$ (e.g., Fig. 6). Eq. 9 thus may lead to an under-estimation in unmeasured R(VOC); however, due to the number of unknowns and uncertainties, it cannot be evaluated at this time."**

[Figure]

**Figure S13. (a) Scatter plot of Eq. 1 versus Eq. 9 $P(O_x)$, colored by $NO_x$ mixing ratios. The slope, 1.24, is red, and the 1:1 line is black. (b) Binned $P(O_x)$ for Eq. 1 (black) and Eq. 9 (blue).**

1.3 Lines 99 ff.: I have some questions regarding the calculations presented in the Supplement:
1.3.1 Line 44 (Supplement) / Eq. S2: What about the reaction of CO with OH? $HO_2$ is formed without going through $RO_2$? Does this need to be accounted for? Depending on the location / altitude, I would expect the $HO_2$ could be up to a factor of 2-3 higher than $RO_2$.

As this is a formulation for urban areas with high VOCs and taken from Farmer et al. (2011) and other publications, to specifically focus on the role of VOCs on OH and ANs, CO was not included in those studies and not included here for consistency. Further, incorporating CO into the equations would be difficult and reduce the simplicity of the equations for illustrative discussions about how $P(HO_x)$, R(VOC), and $NO_x$ sinks control $P(O_x)$. Also, Sect. S1 and this analytical model is to introduce the importance of R(VOC), $P(HO_x)$, and $NO_x$ loss in understanding $P(O_x)$ and not to be over-interpreted in trying to explain the atmosphere 100%.

For the main text, we have included CO, specifically in the weighting of ɣ, etc. (see comment 1.8).

1.3.2 Figure S1b: It would be helpful to show the equation that presents the relationship between $P(O_3)$ and $P(HO_x)$ as well.

$P(HO_x)$ is a constant used to calculate $P(O_x)$. E.g., $P(HO_x)$ is declared as a constant in Eq. S6, and as discussed in line 45 - 46 in the SI, an assumed $P(HO_x)$ is used to calculate [OH] in quadratic formula. Thus, there is no equation to show a relationship as $P(HO_x)$ is necessary to determine $[OH_{Calc}]$ in Eq. S8. Fig. S1b indirectly shows what happens when $P(HO_x)$ (constant variable) increases or decreases.

We have added the following text in the caption for Figure S1 to clarify:

**"Note, for all scenarios/panels here, R(VOC), $P(HO_x)$, and α are constants, as discussed above and shown in Eq. S1 - S8."**

Also, at line 45 in the SI, we have added the following text:

**"Combining Eq. S1 and S2 together with an assumed, constant $P(HO_x)$, . . ."**

1.3.3 Eq. S7 / Figure S1c: Do I understand correctly that Eq. S7 is used as a basis to create Figure S1c? It looks like the $O_3$ production is approximately halved when increasing the branching ratio α from 0 to 10%. However, this is difficult to understand when looking at Eq. S7. The rate constants for $HO_2$ and $RO_2$ with NO are similar (k($HO_2$+NO) is a bit higher), and you assume that $HO_2 \cong RO_2$. Therefore Eq. S7 could be simplified to $P(O_x) \cong (2-α) * k * [HO_2][NO]$. Shouldn't $P(O_x)$ decrease by only a few % for α=0.1? Maybe it could be clarified how Figure S1 is developed / what causes the large impact on $O_3$ production.

Eq. S8 is the equation used to make the final plots in Figure S1. The large decrease in $P(O_x)$ is both due to Eq. S8 as well as Eq. S4 and S5 to calculate OH (Eq. S3). Those instances of α in Eq. S1 - S5 lead to ~40% reduction of OH as well as the 10% reduction of $P(O_x)$ in Eq. S8. We have added the following to line 64 in the SI clarify:

**"Note that α controls both $P(O_x)$ (Eq. S8) and $[OH_{Calc}]$ (Eq. S1, S2, S4, and S5). Thus, reducing α reduces both $[OH_{Calc}]$ by ~40% (going from α = 0.1 to 0.05) and $P(O_x)$ by ~10%."**

1.4 Lines 173 ff.:: Airborne $NO_2$ measurements are a challenge, particularly in the presence of peroxy nitrates, because they decompose in the instrument (where we usually find higher temperatures than those of the ambient air) (Reed et al., 2016; Shah et al., 2023). Usually, this problem arises at higher altitudes, but if you expect large amounts of PNs this might have a bias on the $NO_2$ measurements. Was this investigated? How well does the measured $NO_2$ and the PSS calculated $NO_2$ agree? Maybe a comparison of measured and calculated $NO_2$ beyond the $NO_2$/NO ratio (e.g., in the Supplement) could strengthen your argument.

As discussed in Nault et. al (2015) and Browne et al. (2011), peroxy nitrates, which do not have the stabilization of the acyl group (C=O) for the C-OONO2 bond, have a lifetime less than 1 s for the temperatures and pressures investigated in this study; thus, they would not impact $NO_2$. Pernitric acid may survive at these temperatures in the boundary layer and may potentially be a source of $NO_2$ for either instrument.

We have added the following figure and text to SI S2 at line 97:

**"Finally, photostationary steady-state (PSS) $NO_2$, calculated through rearrangement of Eq. S9, is compared against the measured $NO_2$ by chemiluminescence (CL) and laser induced fluorescence (LIF) in Fig. S4. The measured $NO_{2,LIF}$ versus $NO_{2,PSS}$ is closer to the one-to-one line (slope = 1.06) compared to the measured $NO_{2,CL}$ versus $NO_{2,PSS}$ (slope = 1.23). This further supports the results from Fig. S3, showing that the $NO_{2,LIF}$ ($NO_2$ UCB in Fig. S3) is closer to the predicted PSS $NO_2$."**

[Figure]

**Figure S4. Scatter plot of predicted $NO_2$ PSS, from Eq. S9, and measured $NO_2$, from laser induced fluorescence (LIF) or chemiluminescence (CL). The PSS vs CL slope is 1.23, the PSS vs LIF slope is 1.06, and the 1:1 line is red.**

1.5 Lines 224: Why do you use the box model calculated HO₂ instead of measurements? Maybe you could present a comparison of modeled and measured HO₂.

We have added the following figure and text to the end of SI Section S3 at line 197.

**"As described in Sect. S1 and Eq. 1 (and S7), HO₂ is important in the Oₓ production. However, an intercomparison of measured and F0AM modeled HO₂ shows that the two values diverge from the one-to-one line at high NOₓ mixing ratios (Figure S6a), where the measured HO₂ is higher compared to F0AM modeled HO₂. As this is at high NOₓ mixing ratios, this impacts the calculated P(Oₓ), where the measured HO₂ would suggest high P(Oₓ) with increasing NOₓ whereas the F0AM HO₂ shows decreasing P(Oₓ) with increasing NOₓ (Figure S6b). The latter, decreasing P(Oₓ) with increasing NOₓ, more closely aligns with theory (e.g., Sect. S1 and** (Seinfeld. and Pandis, 2006))**. Further, the latter more closely aligns with observations in that P(Oₓ) increases with decreasing NOₓ, e.g., the "NOₓ penalty"** (Jhun et al., 2015; Pusede and Cohen, 2012)**. Though calculations using observed HO₂ have suggested that P(Oₓ) either remains constant and/or decreases with decreasing NOₓ** (e.g., Whalley et al., 2018, 2021)**, this does not align with both theory and the "NOₓ penalty" observed, suggesting potential uncertainties for HO₂ at low HO₂ and high NOₓ mixing ratios. Thus, to be consistent with theory and "NOₓ penalty" observations, F0AM calculated HO₂ is used throughout the study."**

[Figure]

**Figure S6. (a) Scatter plot of HO₂ predicted from F0AM vs measured HO₂, colored by measured NOₓ mixing ratios. One-to-one line represented by the grey line. (b) Calculated P(Oₓ), using Eq. 1, for HO₂ predicted by F0AM (black) or HO₂ measured (blue).**

1.6 Line 238: Could this also include airport NOₓ emissions?

We have added the following at line 261:

**"The missed approach included low level sampling at a military airport, which may have contributed to the NOₓ mixing ratios along with the activities throughout the SMA."**

1.7 Line 272 ff.: Are these differences significant? What's the uncertainty of the individual shares?

The differences in mixing ratios at each binned value for the different classes both are outside the uncertainty associated with each measurement (maximum 30% uncertainty) and the standard error of the mean, indicating the differences are significant (and may look less significant as fractional contribution).

We have added the following at line 307 to discuss this point:

**"The differences in the binned mean value for each species is greater than the uncertainty associated with its measurements (maximum uncertainty 30%) and greater than the standard error of the mean, indicating that all the percent differences shown here are real."**

1.8 Line 311 f.: Does this mean that one go through the $HO_x$ cycle produces only 1.53, instead of 2 $O_3$? Does this in turn mean that only 1.53 NO molecules are involved? Could you explain the role of CO and HCHO in more detail?

We have clarified $\gamma = 1.53$ in line 338 as:

**"The reactivity weighted $\gamma$ is calculated for the observed and F0AM calculated species with Eq. 11, where $\gamma$ for each compound is taken from MCM** (Jenkin et al., 2015) **and accounts for potential differences in the number of $O_3$ molecules produced per channel per oxidation of VOC (e.g., xylene produces two $O_3$ molecules 60% of the time and one $O_3$ molecule 40% of the time). All the terms were defined for Eq. 8 - 9.**

$$\gamma_{eff} = \frac{\sum_i^{\square} \square \gamma_i k_{OH+VOCi}[OH][VOC_i]}{\sum_i^{\square} \square k_{OH+VOCi}[OH][VOC_i]}$$

**The reactivity weighted $\gamma$ is found to be, on average, 1.53, which is lower than the value of 2 typically assumed in prior studies** (e.g., Perring et al., 2013)**. This lower reactivity weighted $\gamma$ is due to the role of CO ($\gamma = 1$) and $CH_2O$ ($\gamma = 1$) to the total reactivity."**

1.9 Lines 337 - 352: This section is a bit hard to follow. Could you clarify how R(VOC) is determined? Is Eq. 11 needed to understand Figure 4? Maybe it would make sense to present Eq. 11 earlier in the text?

R(VOC) is determine in the following steps: 1) calculated using the observed measurements on the DC-8 with their known rate constants; 2) calculated using the F0AM derived species not observed on the DC-8 to add to the observed measurements on the DC-8; and, 3) calculated to find the necessary unmeasured R(VOC) for the measured R(VOC) to explain the $\Delta O_x/\Delta \Sigma ANs$ slope.

We think the confusion may be with Eq. 11 where we use $RVOC_m$ and $RVOC_u$ as the F0AM species are technically unmeasured; therefore, we have clarified this in line 415 as:

**". . . and $m$ and $u$ correspond to "measured" (measured VOCs on DC-8 along with secondary species predicted by F0AM) and "unmeasured" (unmeasured VOCs that are not represented by DC-8 observations and not predicted by F0AM) RVOC and α."**

Eq. 11 is only needed to understand the red line in Fig. 4a. Introducing it sooner does not flow with the discussion about the observed and modeled R(VOC). We think introducing it sooner without explaining why we derive the equation for this unmeasured (not measured by DC-8 or predicted by F0AM) VOC would lead to confusion.

1.10 Lien 466 ff.: Could you elaborate a bit further on how competition between R8 and R9 relates to formaldehyde?

We have added the following text at line 517:

**"At low $NO$-to-$NO_2$ ratios, R8 is more favorable, as the $R(O)O_2^.$ is more likely to react with $NO_2$ compared with $NO$, leading to more efficient production of PNs over formaldehyde. As $NO$-to-$NO_2$ ratios increase ($NO$ becomes comparable to $NO_2$, leading to more equal probability in $R(O)O_2^.$ reacting to $NO$ and $NO_2$, leading to production of alkoxy radicals that can form formaldehyde), R9 becomes more dominant, leading to less production of PNs."**

1.11 Lines 577 ff.: Are Figures 6(b) and (c) created using the box model or the observations?

We have added the following text to Fig. 6 caption:

**"Also note that F0AM $HO_2$, $CH_3C(O)O_2^.$, $R(O)O_2^.$, and F0AM secondary products are used here along with observations."**

Minor Comments

1.12 Line 84: Is there a word missing? "One important subclass of VOCs *are* (?) aldehydes…"

We have corrected it to say:

**"One important subclass of VOCs are aldehydes (RCHO),. . ."**

1.13 Figure 3: The Figure caption mentions panel (c) instead of (b).

We have corrected it to say (b).

1.14 Line 341 / Figure 4b: Do you mean "α using Eq. 10"?

Yes; it has been corrected (note it is now Eq. 11 as we have added another equation). See comment 2.1.

1.15 Line 568 f.: There seems to be something wrong in this sentence. Can you rephrase it?

We have changed line 628 to say:

**"The $NO_x$ distribution over SMA (Figure 1) shows a large area (~127.53°E to 127.18°E, or ~39 km), which corresponds to the $NO_x$ mixing ratio that results in maximum $P(O_x)$, as shown in Figure 6."**

*Reviewer RC2 (Reviewer #1)*

2.0 Nault et al. present an intriguing dataset which shows that the oxidized NOx budget measured during KORUS-AQ includes alkyl (AN) and peroxy nitrates (PN) that cannot be explained by the emissions and chemistry represented by common chemical mechanisms. The authors use observations to show that a significant source of VOC reactivity [R(VOC)] is needed to explain observation of OH reactivity and potentially close the NOz budget. The authors show that this missing chemistry has an important impact on radical production and loss rates, and thus predictions of ozone formation. The authors assess potential sources and suggest that aldehydes from cooking and other oxygenated VOCs could explain the missing chemistry.

I found the study very interesting and the authors provide number of useful constraints to assess the role of understudied chemistry impacting the air quality in Seoul. I think this is a valuable contribution to characterization of urban air quality. I have a few comments below that I hope will help to strengthen the discussion.

We thank the reviewer for the overall positive review and detailed comments. We have addressed each point below.

Major comments

2.1 Lines 373 - 397: I think the comparison of F0AM R(VOC) to PSU measurements convincingly shows the missing chemistry in the model. In my opinion, the extrapolation to higher NOx using equation 11 is a bit speculative and not really necessary to make the authors point. While I appreciate that there is a lot of discussion about the uncertainty in this approach, I'd suggest leaning into the observations and comparison to previous studies as written at lines 400-410. To my eye, the missing R(VOC) and the modeled VOC distribution are not drastically different at low NOx and high NOx.

We think including the R(VOC) for the full range of $NO_x$ is necessary, specifically as that directly goes into evaluating $P(O_x)$ and $L(HO_x)$ in Fig. 6. Further, we are wanting to show a different method to constrain R(VOC) when direct measurements of R(VOC) are missing or are impacted by interferences, such as high $NO_x$ mixing ratios. There are differences in the amount of unmeasured R(VOC) as $NO_x$ increases, which we have updated Fig. 4 to show.

[Figure]

**Figure 4. (a) Upper panel is the binned calculated (calc.) unmeasured VOC reactivity (R(VOC)ᵤ). Note, unmeasured is for any species not measured on DC-8 or constrained by F0AM and is calculated using Eq. 13. Lower panel is binned VOC reactivity versus NOₓ observed over SMA during KORUS-AQ (see Figure 1 for the area studied). The measured observed R(VOC), labeled as "From PSU", where PSU is Pennsylvania State University, is the VOC reactivity calculated from the measured total OH reactivity with inorganic OH reactivity removed. As discussed in Brune et al.** (2022)**, the OH reactivity has interferences at high NOₓ mixing ratios. The error bar is the uncertainty in the OH reactivity measurement** (Brune et al., 2022)**. The red line represents the calculated unmeasured R(VOC), using Eq. 11, with an assumed α = 0.10. The shaded area represents different calculated unmeasured R(VOC), assuming different α for the unmeasured R(VOC) (see Eq. 11). (b) The calculated effective α from observations versus NOₓ. The dashed purple line is the effective α estimated from Eq. 10, using the slope from Figure 3a. For both (a) and (b), the colored stacked data is the calculated VOC reactivity (a) and weighted effective α (b). The values from (b) are calculated using Eq. 11. Finally, for both (a) and (b), F0AM species is the reactivity for compounds not measured on the DC-8 predicted by F0AM with an estimated α = 0.05. The associated uncertainty in using different α for the F0AM predicted reactivity is explored in Figure S4.**

Further, one potential aspect that may impact and make unmeasured R(VOC) appears small changes with NOₓ is assuming a constant α (as well as constant γ) at all NOₓ mixing ratios. As seen in Fig. 4b, the calculated α appease to be changing with NOₓ due to changes in the R(VOC). As defining Oₓ and ΣANs background is challenging due to the changing meteorological conditions, attempting to constrain α at different NOₓ mixing ratios would be highly uncertain. We have added the following text at line 430 to highlight these limitations:

**"Another limitation in this study is assuming a constant α and γ across all NOₓ mixing ratios to estimate the unmeasured R(VOC). At higher NOₓ mixing ratios, where the VOC mixing ratios would be highest due to being closer to emissions, it would be expected that both α and**

**γ would change. However, direction that these values would change is uncertain as both α and γ depend on the structure of the VOC, which is currently unknown."**

2.2 Discussion of PANs: The authors note that the model overpredicted PAN by a factor of 2 and that this could be related to the assumed background, dilution rates, and/or temperature. Do the authors have a sense of the major cause for this discrepancy? My concern is that if this is mostly affected by temperature, then the higher PNs would also be affected and bias the model/measurement comparison shown in Fig. 5a. Can the authors provide some sensitivity analyses to determine how much each factor might affect net PAN production?

At this state, the potential source of the overprediction in PAN by F0AM is unknown. As such, we have constrained the F0AM model to observed PAN and PPN, as discussed in comment 2.3. Though we are not able to constrain the higher PNs and thus using F0AM model to indicate that the difference in measured and modeled PNs may further collaborate unmeasured R(VOC), the things that may be impacting PAN diel steady-state may have less of an impact for the higher PNs, as they are more thermally stable compared to PAN due to the longer carbon backbone (e.g., Kabir et al., 2014).

We have added the following at line 204 to address this:

**"Note, the reason PAN and PPN were constrained were due to uncertainties in the thermal lifetime, temperature history, and dilution rate used in F0AM, which had larger impacts on the $CH_3C(O)O_2^{\cdot}$ and PAN than on other unconstrained compounds (e.g., OH and formaldehyde and not shown;** Brune et al. (2022)**). Part of this larger impact is due to $CH_3C(O)O_2^{\cdot}$ being one of the most abundant radicals and one of the final radical products in the oxidation of numerous compounds** (e.g., Jenkin et al., 2015)**. We do not expect these uncertainties to impact the higher PNs as (a) there are less precursors to form them compared to PAN and (b) they are expected to have higher thermal stability compared to PAN due to longer carbon backbone** (Kabir et al., 2014)**."**

2.3 I would also like to note that the discrepancy on PAN could also be due to uncertainties in the ethanol constraint. Since ethanol is a large component of the PAN budget (Fig. 5c), it would be worth exploring an emission sensitivity.

We have changed the F0AM modeling in that we constrained PAN and PPN for the calculation of other PNs and the radical species, as we have updated in line 191:

**"We constrain concentrations of $NO$, $O_3$, $H_2O_2$, $HNO_3$, $CO$, $CH_4$, $H_2$, PAN, PPN, and all measured or estimated VOCs given in Table 2 and Table S1 to calculate $HO_2$, all organic peroxy and acyl peroxy radicals, and unmeasured PNs. To calculate the PAN and PNs budget, we allow . . ."**

We have also removed the PAN and PPN comparison between F0AM and observations as we are constraining both of these gases in F0AM, as shown in updated figure below:

[Figure]

**"Figure S9. Evaluation of the F0AM model performance versus gases measured on DC-8 over the SMA and not used to constrain the model. (a) Scatter plot of F0AM predicted $NO_2$ versus the observed $NO_2$ from UC Berkeley. (b) Scatter plot of the F0AM predicted OH versus Penn State observed OH. (c) Scatter plot of F0AM predicted $CH_2O$ versus CAMS observed $CH_2O$."**

Finally, as highlighted in the updated text above, all measured species, which includes acetaldehyde, were constrained. Since ethanol directly forms acetaldehyde, any uncertainty associated with ethanol would not have impacted acetaldehyde.

2.4 Finally, the GT mass spec reports PBzN. since PBzN has a more limited number of precursors, it might be worth comparing the F0AM output to this species. This might help rule out the impact of processes that would affect most PNs (e.g., temperature) from those that are specific to individual PN species (e.g., emissions).

Thank you for the suggestion. However, a co-author reported to us that they think PBzN may be underestimated due to possible inlet losses, as discussed in Zheng et al. (2011). We have added the following at line 179 to clarify this point:

**"Further, though PBzN was measured by GT-CIMS (Table 2), it is not compared with calculated PBzN from F0AM (Sect. 2.2) as it may be underestimated due to possible inlet losses, as discussed in Zheng et al.** (2011)**."**

Minor Comments

2.5 Line 46: This is a bit contrary with sentence 33 - 34 where increases in O3 seem to be attributed to decreases in NOx in a NOx-saturated ozone production regime. Perhaps clearer to say "in altering" O3 production?

We have updated the text at line 46 to say:

**". . . the results here highlight the role of $\Sigma$PNs play in urban environments in altering net O3 production. . ."**

2.6 Line 61: from?

We corrected form to from.

2.7 Line 62: "product" should be plural.

We have made product plural.

2.8 Line 84: add "are" before aldehydes

As discussed in response 1.12, we have updated.

2.9 Line 165: Is there another description for the blocking period other than "blocking conditions"

We have rephrased line 166 to say:

". . ., where a high pressure ridge is located to an area north of lower pressure, which can preclude significant changes in synoptic meteorology and results in occasional stagnant conditions/minimal pollution transport (Peterson et al., 2019)."

2.10 Line 182: For each "aircraft observation" is this 1-min data, or some average of urban plume intercepts? The term "diurnal cycle mode" is unclear. Which species are being evaluated for convergence? Is this meant to be O3, HCHO, or other specific species? And is the model moved forward in time based on the aircraft observations until Obs = Model? I realize this is in Schroeder et al., but some small details here would be useful.

We included further information at line 187:

"As in Schroeder et al. (2020), we simulate each 1-min merged aircraft observation through the full diurnal solar cycle (i.e., diel steady-state), until the diurnal cycle for each unconstrained species reaches convergence within 1%. These unconstrained species, such as formaldehyde, $NO_2$, and OH, are then evaluated to ensure consistency between the F0AM model and aircraft observations."

2.11 Equation 5: The equation represents primary HOx and photolysis of HOx reservoirs. Are there other sources that might matter in this formulation, or is this the majority of species that contribute to HOx production in the model? What about photolysis of higher-peroxides formed by RO2 + HO2? These are treated as a loss of HOx, but wouldn't their photolysis contribute to HOx well?

We have added the following text at line 239 to clarify Eq. 5:

"In Eq. 5, only values directly measured on the DC-8 during KORUS-AQ are included. As discussed in Wang et al. (2022) and Sect. 4.3, this is most likely an underestimation of $P(HO_x)$."

2.12 Ultimately, I'm wondering what fraction of the HOx production term is represented by these measures. My understanding is that this can be derived from F0AM by summing up all the terms that produce OH and HO2 and excluding reactions that interconvert HOx (e.g., NO + HO2).

Also, we have added the following text and analysis at line 702:

**"A comparison of HOₓ sources for F0AM is shown in Figure S14. As it has more complete OVOCs than the observations, the contributions are different than shown in Figure 7. Both observations and F0AM agree that photolysis of O₃ and subsequent reaction with water (R12) and photolysis of formaldehyde are the two largest sources of HOₓ. F0AM also shows that methylglyoxal is an important source of HOₓ, which is not shown in Figure 7 as methylglyoxal was not measured. However, the total F0AM P(HOₓ) was ~2.4 ppbv hr⁻¹, which was lower than the observationally constrained value. This further supports either potential unmeasured OVOCs coming from both emissions and chemistry and/or uncertainty in the photolysis rate constants for these OVOCs** (e.g., Wang et al., 2022)**.**

[Figure]

**Figure S14. Fractional contribution for different sources of HOₓ predicted from F0AM."**

2.13 Line 224-225: This seems out of place without also knowing that HO2 was measured onboard the aircraft. Perhaps clarify by noting that "HO2 calculated from F0AM, rather than aircraft measurements (Crawford et al., 2021), is used in the equation to determine the Ox and HOx budget." Is there a reason for using modeled HO2 as opposed to observations?

We have updated line 244 to say:

**"Finally, HO₂ calculated from F0AM, rather than aircraft measurements** (Crawford et al., 2021)**, is used in the equations to determine the Oₓ and HOₓ budget (see Sect. S3, Figure S6)."**

See comment 1.5 about why F0AM $HO_2$ was used.

2.14 Line 264:  Are there studies other than Wooldridge that also show closure? It seems that the number of urban sites reported in that study are limited, and the disagreement observed in the key urban study (PIE in Boulder, CO) is attributed to poor inlet design. It would be helpful if other studies were cited here.

We agree this would be helpful but unfortunately, to the best of our knowledge, there are no other studies after Wooldridge et al. (2010) that had total PNs measured by thermal dissociation while concurrently measuring individual PNs by mass spectrometry.

2.16 Lines 274-275:  At any point, do the measurements (either in NOy or the sum of individual NOz components) begin to reach detection limits? May be useful to note which measurements are still above detection limit at this point?

To the best of our knowledge, none of the data is near detection limits as unreported data could be due to detection limits, calibrations, or reasons (e.g., instrumental maintenance).

2.17 Lines 311 - 312:  How exactly is y calculated? Is this a model simulation (e.g., the yield of ozone over the full oxidation of the molecule) or is this inferred from the mechanism branching ratio? And how is this weighted for every VOC? Some details here (perhaps with an equation) would be helpful to understand this calculation.

See response 1.8

2.18 Line 326:  I presume that the units of the Ox/sumANs is ppb / ppb?

Yes. We have included these units in text and in the caption for Fig. 3.

2.19 Line 337 - 339:  It would be helpful to see an equation for how $\alpha_{eff}$ is calculated from the observations. It seems the authors are not including the intermediates in the calculation of $\alpha_{eff}$. Are the F0AM intermediates a negligible component?

We have added the following text to clarify at line 369:

"..., Eq. 12 was used, where all the terms are the same as Eq. 8 - 9.
$$\alpha_{eff} = \frac{\sum_i^{\square} \square \alpha_i k_{OH+VOCi}[OH][VOC_i]}{\sum_i^{\square} \square \gamma_i k_{OH+VOCi}[OH][VOC_i]} \qquad (12)"$$

The F0AM intermediates are included and are an important fraction of $\alpha_{eff}$, as shown in Fig. 4b as the purple values that are labeled F0AM species (see comment 2.1 for Fig. 4b).

2.20 Line 415:  Cooking emissions are rich in a suite of long-chain aldehydes (e.g., Schauer et al., 2002) and so it may be better to discuss those emissions as a group. Coggon et al. (2024) recently showed that C2 - C11 aldehydes from cooking are present in ambient air but not well characterized

in ambient datasets. It would be more encompassing to say "one possible group of missing VOCs are long-chain aldehydes from cooking and vegetative emissions, including nonanal."

We have corrected line 460 to say this.

2.21 Line 418-419: Did the PTR observe nonanal? It is possible that the interference could be significantly impacted by a C5-aldehyde, which almost completely undergoes fragmentation to produce the isoprene signal (Buhr et al., 2002). Thus, if nonanal is not super abundant, other aldehydes (or cycloalkanes) could be contributing and not easily detected at the proton-transfer product.

We have updated line 464 to say:

**"Higher carbon aldehydes (or cycloalkanes) have been recently suggested to be a potential interference with isoprene measurements on a PTR-MS."**

2.22 Line 430: I suggest rewriting to say "nonanal and other long-chain aldehydes" may be an important PN precursors.

We have corrected line 476 to say this.

2.23 Line 457: The oVOCs discussed up until this point were mostly associated with aldehydes and emissions with cooking. What about solvent sources, such as VCPs? According to McDonald et al. (2018), these emissions may contain significant contributions from glycols and alcohols, which aren't very well measured by PTR or GC.

We have updated line 479 to more explicitly include solvents as well as cooking:

**"OVOC emissions from multiple sources, including solvent evaporation and other non-transportation emissions, are generally considered to be an important fraction of R(VOC) for urban emissions but may not be measurable by PTR or GC, such as glycols** (Gkatzelis et al., 2021; de Gouw et al., 2018; Ma et al., 2022; McDonald et al., 2018; Simpson et al., 2020; Wang et al., 2022; Yang et al., 2022)**."**

2.24 Lines 481 - 486: I was confused by this section. At line 481, it reads as though the authors are calculating higher PNs from the model by subtracting PAN, but then in the following sentences it is noted that PAN was excluded from this analysis. Could this be clarified?

We have corrected lines 534 to say:

**"This missing R(VOC) is further observed in the F0AM-predicted higher PNs ($\Sigma$PNs without PAN, or $\Sigma$PNs-PAN for short) versus formaldehyde, . . ."**

2.25 Line 514: PBzN is reported by the GT CIMS. Can these species be compared?

We cannot, as discussed in comment 2.4.

2.26 Line 515: It would be worth noting that PHAN formation may be overestimated in the MCM. Butkovskaya et al. (2006) present evidence showing that the RO2 radical formed from glycolaldehyde + OH (i.e., the PHAN precursor) decomposes to formaldehyde and CO2. This pathway could compete with other RO2 pathways and limit PHAN production. Other studies (e.g., Magneron et al., 2005) note that PN's were not observed in glycolaldehyde + OH oxidation, and this decomposition pathway could be a possible cause.

We have added the following information at line 566:

**"Note, PHAN formation in MCM/F0AM may be overestimated, as Butkovskaya et al.** (2006) **found that the radical formed from the photooxidation of glycolaldehyde decomposes to form formaldehyde and CO₂, potentially competing with the pathway to form PHAN. Other studies also found that PNs were not observed by the photooxidation of glycolaldehyde** (Magneron et al., 2005)**."**

2.27 Lines 535 - 547: Ethanol is in most VCPs. I would expand this statement about cleaning agents to include all VCP sectors.

We have updated line 593 to say:

**"Ethanol is considered to mainly come from both vehicle emissions** (e.g., Millet et al., 2012) **and non-transportation emissions, including cleaning agents and solvents** (e.g., McDonald et al., 2018)**."**

2.28 Line 592 - 594: Please point towards Fig 6c in this sentence.

We have included this (line 654).

2.29 Figure 5: The legend in Fig 5b is confusing. Some species refer to PNs while others refer to precursors of PNs, which I presume are then lumped together for that specific precursor (correct?). Perhaps clear to say "monoterpene-derived PNs, isoprene derived PNs..." or some acronym (MONOPN, ISOPN...etc).

We have updated the legend, as shown below. Note, Fig. 5a has also been updated as we are only using constrained PAN except for the budget analysis.

[Figure]

**Figure 5. (a)** Scatter plot of binned higher $\Sigma$PNs calculated using F0AM (red) or binned higher $\Sigma$PNs from observations (black) versus formaldehyde (CH₂O). Slopes shown are ODR fits to the binned data. PPN and PAN were constrained by observations for F0AM while all the other higher PNs were not constrained **(b)** Fractional contribution of the higher PNs predicted from F0AM versus NOₓ. **(c)** Fractional contribution of different precursors to PAN, predicted by F0AM versus NOₓ. For both (b) and (c), Alk is all alkanes, Arom is all aromatics, and ≥C₄ Alk is all alkanes with 4 or more carbons. See Figure S8 for comparison of F0AM.

Separate from both reviews. We found one error in our calculation of net P(Oₓ) for Figure 6. We have corrected it and Figure 6. It reduces the max P(Oₓ) from 10.3 ppbv hr⁻¹ to 9.3 ppbv hr⁻¹. We have updated the text within the paper to reflect that.

---

## Author Response (AR2)

**1 Using observed urban NOx sinks to constrain VOC reactivity and the ozone and radical**

**2 budget in the Seoul Metropolitan Area**

- 3 Benjamin A. Nault1,2,\*, Katherine R. Travis3, James H. Crawford3, Donald R. Blake4, Pedro
- 4 Campuzano-Jost5, Ronald C. Cohen6, Joshua P. DiGangi3, Glenn S. Diskin3, Samuel R. Hall7, L.
- 5 Gregory Huey8, Jose L. Jimenez5, Kyung-Eun Kim9, Young Ro Lee8,a, Isobel J. Simpson4, Kirk
- 6 Ullmann7, Armin Wisthaler10,11
- 7 1CACC, Aerodyne Research, Inc., Billerica, MA, USA

[revised manuscript text omitted]
_3C(O)CH_3C(O)CH_3C(O)CH_3C(O)CH_3C(O)CH_3C(O)CH_3C(O)CH_3C(O)CH_3C(O)CH_3C(O)CH_3C(O)CH_3C(O)CH_3C(O)CH_3C(O)CH_3C(O)CH_3C(O)CH_3C(O)CH_3C(O)CH_3C(O)CH_3C(O)CH_3C(O)CH_3C(O)CH_3C(O)CH_3C(O)CH_3C(O)CH_3C(O)CH_3C(O)CH_3C(O)CH_3C(O)CH_3C(O)CH_3C(O)CH_3C(O)CH_3C(O)CH_3C(O)CH_3C(O)CH_3C(O)CH_3C(O)CH_3C(O)CH_3C(O)CH_3C(O)CH_3C(O)CH_3C(O)CH_3C(O)CH_3C(O)CH_3C(O)CH_3C(O)CH_3C(O)CH_3C(O)CH_3C(O)CH_3C(O)CH_3C(O)CH_3C(O)CH_3C(O)CH_3C(O)CH_3C(O)CH_3C(O)CH_3C(O)CH_3C(O)CH_3C(O)CH_3C(O)CH_3C(O)CH_3C(O)CH_3C(O)CH_3C(O)CH_3C(O)CH_3C(O)CH_3C(O)CH_3C(O)CH_3C(O)CH_3C(O)CH_3C(O)CH_3C(O)CH_3C(O)CH_3C(O)CH_3C(O)CH_3C(O)CH_3C(O)CH_3C(O)CH_3C(O)CH_3C(O)CH_3C(O)CH_3C(O)CH_3C(O)CH_3C(O)CH_3C(O)CH_3C(O)CH_3C(O)CH_3C(O)CH_3C(O)CH_3C(O)CH_3C(O)CH_3C(O)CH_3C(O)CH_3C(O)CH_3C(O)CH_3C(O)CH_3C(O)CH_3C(O)CH_3C(O)CH_3C(O)CH_3C(O)CH_3C(O)CH_3C(O)CH_3C(O)CH_3C(O)CH_3C(O)CH_3C(O)CH_3C(O)CH_3C(O)CH_3C(O)CH_3C(O)CH_3C(O)CH_3C(O)CH_3C(O)CH_3C(O)CH_3C(O)CH_3C(O)C(O)CH_3C(O)C(O)CH_3C(O)$                                                                                                                                                                                                                 |                         |
| 231 | $2j_{CH_3CH_2C(O)CH_3}[CH_3CH_2C(O)CH_2]$                                                                                                                                                                                                                                                                                                                                                                                                                                                                                                                                                                                                                                                                                                                                                                                                                                                                                                                                                                                                                                                                                                                                                                                                                                                                                                                                                                                                                                                                                                                                                                                                                                                                                                                                                                                                                                                                                                                                                                                                                                                                                                                                                                                                                                                                                                      | (5)                     |
| 232 | $L(HO_{x}) = k_{NO_{2}+OH}[NO_{2}][OH] + \sum_{i} \alpha_{eff} k_{RO_{2},i} + NO[RO_{2},i][NO] + NO[RO_{2},i][NO$ |                         |
| 233 | $2k_{HO_2+HO_2}[HO_2][HO_2] + 2k_{RO_2+RO_2}[RO_2][RO_2] + 2k_{HO_2+RO_2}[HO_2][RO_2] + net(PNs)$                                                                                                                                                                                                                                                                                                                                                                                                                                                                                                                                                                                                                                                                                                                                                                                                                                                                                                                                                                                                                                                                                                                                                                                                                                                                                                                                                                                                                                                                                                                                                                                                                                                                                                                                                                                                                                                                                                                                                                                                                                                                                                                                                                                                                                              | (6)                     |
| 234 | $[\mathrm{RO}_{2}^{\cdot}] = \frac{\sum_{i  k_{\mathrm{OH+VOC},i}} [\mathrm{VOC}_{i}] [\mathrm{OH}]}{(1 - \alpha_{\mathrm{eff}}) k_{\mathrm{RO}_{2} + \mathrm{NO}} [\mathrm{NO}] + k_{\mathrm{RO}_{2} + \mathrm{HO}_{2}} [\mathrm{HO}_{2}]}$                                                                                                                                                                                                                                                                                                                                                                                                                                                                                                                                                                                                                                                                                                                                                                                                                                                                                                                                                                                                                                                                                                                                                                                                                                                                                                                                                                                                                                                                                                                                                                                                                                                                                                                                                                                                                                                                                                                                                                                                                                                                                                   | (7)                     |
| 235 | Here, k is the rate constant for compound, i, with the associated compound listed, o                                                                                                                                                                                                                                                                                                                                                                                                                                                                                                                                                                                                                                                                                                                                                                                                                                                                                                                                                                                                                                                                                                                                                                                                                                                                                                                                                                                                                                                                                                                                                                                                                                                                                                                                                                                                                                                                                                                                                                                                                                                                                                                                                                                                                                                           | a eff is the |
| 236 | effective branching ratio for R2a and R2b for the observations (Sect. 3.2), f is the fraction                                                                                                                                                                                                                                                                                                                                                                                                                                                                                                                                                                                                                                                                                                                                                                                                                                                                                                                                                                                                                                                                                                                                                                                                                                                                                                                                                                                                                                                                                                                                                                                                                                                                                                                                                                                                                                                                                                                                                                                                                                                                                                                                                                                                                                                  | that O 1 D   |
| 237 | that reacts with water to form OH versus reacting with a third body molecule to form $O^{3}$                                                                                                                                                                                                                                                                                                                                                                                                                                                                                                                                                                                                                                                                                                                                                                                                                                                                                                                                                                                                                                                                                                                                                                                                                                                                                                                                                                                                                                                                                                                                                                                                                                                                                                                                                                                                                                                                                                                                                                                                                                                                                                                                                                                                                                                   | P, $\beta$ is the       |

quantities (mixing ratios and photolysis rates, Table 2), results from FOAM (Sect. 2.2), estimated

218

238

the

fraction the R(O)O2- that reacts with NO2 versus NO, and j is the measured photolysis frequency

**11**

239 (Table 2). In Eq. 5, only directly values directly measured on the DC-8 during KORUS-AO are 240 included. As discussed in Wang et al. (2022) and Sect. 4.3,, this is most likely an underestimations 241 of  $P(HO_x)$ . Note,  $R(O)O_2$  is not included in Eq. 7 as (a) it is assumed the initial production of 242  $R(O)O_2$  is captured with the reaction of OH with VOC and (b)  $R(O)O_2$  accounts for a small 243 fraction of the total RO2 (< 10%). Not including R(O)O2 in Eq. 7 may lead to a small 244 underestimation of total RO2. Finally, HO2 calculated from F0AM, rather than aircraft 245 measurements (Crawford et al., 2021), is used in the equations to determine the  $O_x$  and  $HO_x$  budget 246 (see Sect. S3, Figure S6).

247

**248 3. Observational constraints on NOx organic oxidation chemistry**

In the Sect. 3.1, the detailed observations from the DC-8 during KORUS-AQ provided measurements that allow us to test our understanding of NOx oxidation into total NOz (NOz = higher NOx oxides, including  $\Sigma$ PNs,  $\Sigma$ ANs, HNO3 and particulate nitrate, pNO3), 
[revised manuscript text omitted]
 $\gamma$ is calculated for the observed and FOAM calculated                     |   |
| 339 | compounds species from F0AM using the values from MCM with Eq. 10, where $\gamma$ for each                                                 |   |
| 340 | compound is taken from MCM (Jenkin et al., 2015) and accounts for potential for difference                                                 |   |
| 341 | number of O 3 molecules produced per channel per oxidation (e.g., xylene produces two O 3                            |   |
| 342 | molecules 60% of the time and one O 3 molecule 40% of the time). All the terms were defined for                                 |   |
| 343 | Eq. 8 – 9.                                                                                                                          |   |
| 344 | $\gamma_{eff} = \frac{\sum_{i} \gamma_i k_{OH+VOC_i} [OH] [VOC_i]}{\sum_{i} k_{OH+VOC_i} [OH] [VOC_i]} $ (10)                              | / |
| 345 | The reactivity weighted <math>\gamma_{\tau}</math> is found to be, on average, 1.53, which is lower than the value of 2             |   |
|     |                                                                                                                                            |   |

The reactivity weighted  $\gamma_7$  is found to be, on average, 1.53, which is lower than the value of 2 typically assumed in prior studies (e.g., Perring et al., 2013). This lower reactivity weighted  $\gamma$  is due to the role of CO ( $\gamma = 1$ ) and CH2O ( $\gamma = 1$ ) to the total reactivity. After the boundary layer height has stabilized (e.g., after 11:00 am LT used here) and is near enough (e.g., less than 1 day aging) to the VOC source to ignore deposition and entrainment, Eq. 8 and 9 can be combined to approximate the change in Ox per molecule  $\Sigma$ AN formed:

$$\frac{\Delta O_x}{\Delta \Sigma A N_S} \approx \frac{P_{O_x}}{P \Sigma A N_S} \approx \frac{1.53(1-\alpha)}{\alpha} \tag{110}$$

For this equation to be valid,  $\alpha$  needs to be relatively small ( $\alpha \ll 1$ ), which is true for VOCs, as maximum  $\alpha$  
[revised manuscript text omitted]
(HOx) while net R8 (sum of higher  $\Sigma$ PNs 658 and PAN) comprises 30 - 40% of L(HOx). At lower NOx mixing ratios, R11 is always smaller for 659  $L(HO_x)$  than net R8, where R11 is about a factor of 2 lower than net R8. Production of  $\Sigma ANs$ 660 played a minor role due to the low  $\alpha_{eff}$ .

The self-reaction of HOx species (R15 – R16) contributes minimally to  $L(HO_x)$  (less than 10%) for NOx mixing ratios greater than 8 ppbv. At lower NOx mixing ratios, R16 starts dominating  $L(HO_x)$  budget, increasing from 8% at 8 ppbv to 50% of  $L(HO_x)$  at NOx mixing ratios less than 2 ppbv. Reaction R15 remains relatively small for the  $L(HO_x)$  budget, only reaching 7% of the  $L(HO_x)$  budget at NOx mixing ratios less than 2 ppbv.

666

**667 4.3 Sources of HOx over SMA**

[revised manuscript text omitted]